# WGFormer: An SE(3)-Transformer Driven by Wasserstein Gradient Flows for Molecular Ground-State Conformation Prediction

**Fanmeng Wang** [1]   **Minjie Cheng** [1]   **Hongteng Xu** [1 2 3]

## Abstract

Predicting molecular ground-state conformation (i.e., energy-minimized conformation) is crucial for many chemical applications such as molecular docking and property prediction. Classic energy-based simulation is time-consuming when solving this problem, while existing learning-based methods have advantages in computational efficiency but sacrifice accuracy and interpretability. In this work, we propose a novel and effective method to bridge the energy-based simulation and the learning-based strategy, which designs and learns a Wasserstein gradient flow-driven SE(3)-Transformer, called WGFormer, for ground-state conformation prediction. Specifically, our method tackles this task within an auto-encoding framework, which encodes low-quality conformations by the proposed WGFormer and decodes corresponding ground-state conformations by an MLP. The architecture of WGFormer corresponds to Wasserstein gradient flows — it optimizes conformations by minimizing an energy function defined on the latent mixture models of atoms, thereby significantly improving performance and interpretability. Extensive experiments demonstrate that our method consistently outperforms state-of-the-art competitors, providing a new and insightful paradigm to predict ground-state conformation. The code is available at https://github.com/FanmengWang/WGFormer.

## 1. Introduction

Molecular ground-state conformation represents the most stable 3D molecular structure corresponding to the energy-minimized state on the potential energy surface. It determines many important molecular properties (Xu et al., 2023) and thus plays a crucial role in various downstream applications like molecular docking (Chatterjee et al., 2023; Paggi et al., 2024) and molecular property prediction (Moon et al., 2023; Liu et al., 2024). Traditionally, molecular ground-state conformation can be obtained through various energy-based simulation methods, e.g., molecular dynamics (MD) simulation (Hollingsworth & Dror, 2018) and density functional theory (DFT) calculation (Pracht et al., 2020). However, these methods are too expensive and computationally slow to meet the growing enormous demands (Wang et al., 2023a; Cen et al., 2024; Li et al., 2025a). Although some cheminformatic tools like RDKit (Landrum et al., 2013) have been developed to generate molecular conformations efficiently, they suffer from undesired precision when obtaining ground-state conformations.

Recently, with the significant advancement of artificial intelligence in the scientific field (Wang et al., 2023b; Abramson et al., 2024), learning-based methods have emerged as a promising solution to tackle this task (Kim et al., 2025). Early methods like ConfVAE (Xu et al., 2021b) employ various generative models to generate multiple potential low-energy conformations and then still require screening the energy-minimized conformation among them, which lack the guarantee of obtaining ground-state conformations due to the uncertainty of the two-stage process (Xu et al., 2023). In this context, subsequent efforts have shifted to design specialized methods to predict molecular ground-state conformation directly (Xu et al., 2021d). Among them, some state-of-the-art methods, e.g., ConfOpt (Guan et al., 2022), formulate ground-state conformation prediction as an optimization problem, which takes low-quality conformations as input and optimizes these conformations accordingly. Essentially, these methods aim to apply neural networks to predict the gradient field of the conformational energy landscape and thus imitate gradient descent to update the input conformations. However, their neural networks are designed empirically, making the feedforward computations generally not correspond to minimizing a physically meaningful energy function of conformations. Therefore, such empirical neural network architectures not only harm the model interpretability but also lead to sub-optimal performance.

[1]Gaoling School of Artificial Intelligence, Renmin University of China [2]Beijing Key Laboratory of Research on Large Models and Intelligent Governance [3]Engineering Research Center of Next-Generation Intelligent Search and Recommendation, MOE. Correspondence to: Hongteng Xu <hongtengxu@ruc.edu.cn>.

*Proceedings of the 42$^{nd}$ International Conference on Machine Learning*, Vancouver, Canada. PMLR 267, 2025. Copyright 2025 by the author(s).

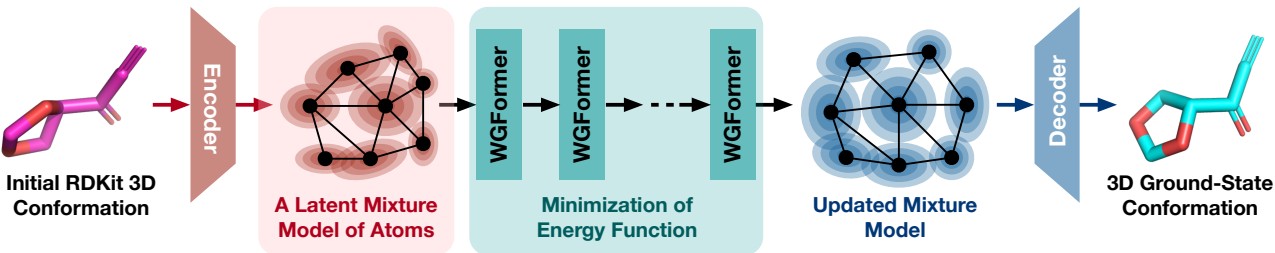

*Figure 1.* An illustration of the proposed model architecture, in which WGFormer corresponds to Wasserstein gradient flows and minimizes a physically reasonable energy function defined on the latent mixture models of atoms.

In this work, considering that low-quality conformations can be readily available (Zhou et al., 2022; Feng et al., 2024), and specialized model architectures designed for 3D molecules have made great progress (Fuchs et al., 2020; Satorras et al., 2021), we design and learn a Wasserstein gradient flow-driven SE(3)-Transformer, called WGFormer, to obtain molecular ground-state conformation through optimizing the corresponding low-quality conformation. As illustrated in Figure 1, our method tackles this task within an auto-encoding framework, where we first encode low-quality conformations (i.e., those generated by RDKit) through our WGFormer, and then decode the corresponding ground-state conformations through a simple multi-layer perceptron (MLP). Inspired by the Sinkformer in (Sander et al., 2022), WGFormer adjusts the query, key, and value matrices and applies the Sinkhorn-scaling algorithm (Cuturi, 2013) to compute attention maps that involve relational information. Such an architecture makes our WGFormer work as Wasserstein gradient flows (Santambrogio, 2017) under specific conditions, which minimizes an energy function defined on the latent mixture models of atoms, significantly improving performance and interpretability.

To the best of our knowledge, our method makes the first attempt to predict molecular ground-state conformation through the lens of Wasserstein gradient flows, which leads to a new and interpretable model architecture for molecular modeling. Extensive experiments consistently demonstrate that our method significantly outperforms state-of-the-art methods across various datasets, achieving promising molecular ground-state conformation prediction results. Meanwhile, comprehensive ablation studies and analyses are also conducted to offer valuable insights into our method, thus validating its rationality and superiority.

## 2. Related Work

### 2.1. Molecular Conformation Generation

In the past few years, many molecular conformation generation methods have been developed (Han et al., 2024; Wang et al., 2025). Early methods like CVGAE (Mansimov et al., 2019) apply variational autoencoders (Kingma & Welling,

2014) to generate conformations. CGCF (Xu et al., 2021a) and ConfGF (Shi et al., 2021) are further proposed to enhance performance through normalizing flows (Kobyzev et al., 2020) and score-based generative models (Song et al., 2020), respectively. Recently, some methods like GeoDiff (Xu et al., 2021c) and TorsionDiff (Jing et al., 2022) leverage popular diffusion models (Ho et al., 2020) to generate conformations. However, these methods are essentially designed to generate multiple "reasonable" conformations rather than directly obtaining ground-state conformations.

Focusing on molecular ground-state conformation prediction, the work in (Xu et al., 2021d) first proposes the Molecule3D benchmark with ground-state conformations determined through density functional theory (DFT) calculations and develops a graph neural network-based solution. GTMGC (Xu et al., 2023) formally defines this task and predicts molecular ground-state conformation based on Graph Transformer (Ying et al., 2021). REBIND (Kim et al., 2025) improves GTMGC by adding edges to molecular graphs based on the Lennard-Jones potential, which captures non-bonded interactions for low-degree atoms. However, these methods predict ground-state conformations merely based on 2D molecular graphs, whose input information may be insufficient to obtain ground-state conformations.

Recently, some methods like ConfOpt (Guan et al., 2022) formulate molecular ground-state conformation prediction from the perspective of conformation optimization. They take low-quality conformations as input and apply various Transformer-based models to predict the gradient field of the conformational energy landscape (Tsypin et al., 2024). Accordingly, the feedforward steps of their models correspond to optimizing the input conformation by gradient descent. However, the empirical architectures of the models prevent them from minimizing a physically meaningful energy function, thus resulting in limited model interpretability and sub-optimal performance. Although some modified Transformer-based models have been proposed from the perspective of differential equations (Dutta et al., 2021; Sander et al., 2022), they are seldom applied to model structured data like molecular conformations and often ignore the requirement of SE(3)-equivariance (Li et al., 2025b).

## 2.2. Applications of Ground-State Conformation

Since molecular ground-state conformation represents the energy-minimized state on the potential energy surface, where the inter-atomic forces are balanced at equilibrium (Kim et al., 2025), it plays a crucial role in various downstream applications (Xu et al., 2023). For example, considering many physical, chemical, and biological properties of a molecule are determined by its ground-state conformation, these conformations are extensively utilized in molecular property prediction (Moon et al., 2023; Liu et al., 2024) and molecular docking (Chatterjee et al., 2023; Paggi et al., 2024). Furthermore, while 3D molecular pretraining has greatly enhanced the performance of various molecule-related tasks (Zhou et al., 2022; Wang et al., 2024), it has also dramatically increased the demand for high-quality 3D molecular structure data, underscoring the critical need to develop accurate and efficient methods for obtaining molecular ground-state conformations.

## 3. Proposed Model

### 3.1. An Auto-Encoding Framework for Ground-State Conformation Prediction

We denote the molecular graph with $N$ atoms as $\mathcal{G} = (\mathcal{V}, \mathcal{E})$, where $\mathcal{V} = \{v_i\}_{i=1}^{N}$ is the set of atoms and $\mathcal{E} = \{e_{ij}\}_{i,j=1}^{N}$ is the set of edges representing interatomic bonds. For the corresponding ground-state conformation, each atom in $\mathcal{V}$ is embedded by a 3D coordinate vector $\boldsymbol{c} \in \mathbb{R}^3$ and the ground-state conformation can be represented as $\boldsymbol{C} = [\boldsymbol{c}_i] \in \mathbb{R}^{N \times 3}$. In addition, the interatomic distances are further denoted as $\boldsymbol{D} = [d_{ij}] \in \mathbb{R}^{N \times N}$, where $d_{ij} = \|\boldsymbol{c}_i - \boldsymbol{c}_j\|_2$ is the Euclidean distance between the $i$-th and $j$-th atom.

Following previous works (Yang et al., 2024; Feng et al., 2024), we apply the cheminformatic tool RDKit (Landrum et al., 2013) to obtain initial low-quality conformation for the molecular graph $\mathcal{G}$, denoted as $\widetilde{C} = [\tilde{\boldsymbol{c}}_i] \in \mathbb{R}^{N \times 3}$. Given the graph structure $\mathcal{G}$ and the initial conformation $\widetilde{C}$, we aim to design and learn a model to predict the ground-state conformation, i.e., $\widehat{C} = g_\theta(\mathcal{G}, \widetilde{C})$, where $\widehat{C} = [\hat{\boldsymbol{c}}_i] \in \mathbb{R}^{N \times 3}$ is the predicted ground-state conformation and $g_\theta$ is the target model with learnable parameter $\theta$. In this work, we tackle this task from the perspective of conformation optimization, designing $g_\theta$ in an auto-encoding framework.

**Encoder.** Following the pipeline of Uni-Mol (Zhou et al., 2022), we first extract the initial atom-level representation $\boldsymbol{X}^{(0)} = [\boldsymbol{x}_i^{(0)}] \in \mathbb{R}^{N \times D}$ and interatomic relational representation $\boldsymbol{R}^{(0)} = [\boldsymbol{r}_{ij}^{(0)}] \in \mathbb{R}^{N \times N \times H}$ from $\mathcal{G}$ and $\widetilde{C}$, respectively. For $i, j = 1, ..., N$, we have

$$\boldsymbol{x}_i^{(0)} = f(v_i), \quad \boldsymbol{r}_{ij}^{(0)} = \mathcal{N}(\tilde{d}_{ij}\boldsymbol{u}_{v_iv_j} + \boldsymbol{v}_{v_iv_j}; \boldsymbol{\mu}, \boldsymbol{\sigma}), \quad (1)$$

where $f$ embeds each atom type $v_i$ to a $D$-dimensional

vector $\boldsymbol{x}_i^{(0)}$, $\tilde{d}_{ij} = \|\tilde{\boldsymbol{c}}_i - \tilde{\boldsymbol{c}}_j\|_2$ is the Euclidean distance between the $i$-th and $j$-th atom within the initial low-quality conformation, and $\mathcal{N}(\cdot; \boldsymbol{\mu}, \boldsymbol{\sigma})$ is a Gaussian kernel function (Scholkopf et al., 1997) with learnable mean value $\boldsymbol{\mu} \in \mathbb{R}^H$ and standard deviation $\boldsymbol{\sigma} \in \mathbb{R}^H$ while $\boldsymbol{u}_{v_iv_j} \in \mathbb{R}^H$ and $\boldsymbol{v}_{v_iv_j} \in \mathbb{R}^H$ represent the learnable weight and bias for each atom type pair $(v_i, v_j)$. Note that, different from Uni-Mol, we ensure that $\boldsymbol{u}_{v_iv_j} = \boldsymbol{u}_{v_jv_i}$ and $\boldsymbol{v}_{v_iv_j} = \boldsymbol{v}_{v_jv_i}$, leading to symmetric relational representation.

Then, we derive the final atom-level and relational representations $\{\boldsymbol{X}^{(L)}, \boldsymbol{R}^{(L)}\}$ by passing the initial atom-level and relational representations $\{\boldsymbol{X}^{(0)}, \boldsymbol{R}^{(0)}\}$ through $L$ proposed WGFormer layers, i.e.,

$$\boldsymbol{X}^{(L)}, \boldsymbol{R}^{(L)} = \text{WGFormer}_L(\boldsymbol{X}^{(0)}, \boldsymbol{R}^{(0)}), \quad (2)$$

where $\text{WGFormer}_L$ refers to the architecture formed by stacking $L$ WGFormer layers together.

**Decoder.** Given the initial relational representation $\boldsymbol{R}^{(0)} = [\boldsymbol{r}_{ij}^{(0)}]$, the final relational representation $\boldsymbol{R}^{(L)} = [\boldsymbol{r}_{ij}^{(L)}]$, and the initial low-quality conformation $\widetilde{C} = [\tilde{\boldsymbol{c}}_i]$, we design an MLP-based decoder to obtain the predicted ground-state conformation $\widehat{C} = [\hat{\boldsymbol{c}}_i]$. For $i = 1, ..., N$, we have

$$\hat{\boldsymbol{c}}_i = \tilde{\boldsymbol{c}}_i + \sum_{j=1}^{N} \frac{\text{MLP}(\boldsymbol{r}_{ij}^{(L)} - \boldsymbol{r}_{ij}^{(0)})(\tilde{\boldsymbol{c}}_i - \tilde{\boldsymbol{c}}_j)}{N}, \quad (3)$$

where $\hat{\boldsymbol{c}}_i \in \mathbb{R}^3$ is the 3D coordinate of the $i$-th atom in the predicted ground-state conformation and $\tilde{\boldsymbol{c}}_i \in \mathbb{R}^3$ is the 3D coordinate of the $i$-th atom in the initial low-quality conformation. The proposed decoder predicts the residual between $\hat{\boldsymbol{c}}_i$ and $\tilde{\boldsymbol{c}}_i$ by the weighted average of the displacement vectors from $\tilde{\boldsymbol{c}}_i$ to the other 3D coordinates (i.e., $\{\tilde{\boldsymbol{c}}_i - \tilde{\boldsymbol{c}}_j\}_{j=1}^{N}$). The weights of the displacement vectors are determined by a simple multi-layer perception (MLP), which takes the residual between $\boldsymbol{r}_{ij}^{(L)}$ and $\boldsymbol{r}_{ij}^{(0)}$ as input. In addition, combined with Eq. (1), we can easily prove that our whole auto-encoding framework is SE(3)-equivariant, and the detailed proof is provided in Appendix A.

The proposed WGFormer plays a central role in our model. Therefore, in the following content, we will introduce its architecture in Section 3.2 and further interpret it from the perspective of Wasserstein gradient flows in Section 4.

### 3.2. The Architecture of WGFormer

Inspired by the Sinkformer in (Sander et al., 2022), the proposed WGFormer implements attention maps based on the Sinkhorn-scaling algorithm (Cuturi, 2013). Consider the WGFormer in the $l$-th layer, which takes $\{\boldsymbol{X}^{(l-1)}, \boldsymbol{R}^{(l-1)}\}$ as input and outputs $\{\boldsymbol{X}^{(l)}, \boldsymbol{R}^{(l)}\}$ accordingly. Specifically, it contains $H$ attention heads, each of which corresponds to a channel of the tensor $\boldsymbol{R}^{(l-1)}$, i.e., $\boldsymbol{R}^{(l-1,h)} \in \mathbb{R}^{N \times N}$ for

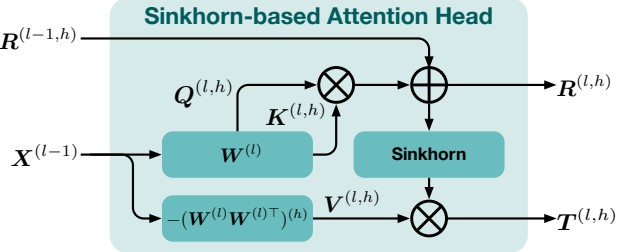

*Figure 2.* An illustration of the $h$-th head in the $l$-th WGFormer.

$h = 1, ..., H$. As illustrated in Figure 2, the attention head is implemented by the following five steps:

$$
\begin{aligned}
&1)\ \boldsymbol{Q}^{(l,h)} = \boldsymbol{K}^{(l,h)} = \boldsymbol{X}^{(l-1)}\boldsymbol{W}^{(l,h)}, \\
&2)\ \boldsymbol{A}^{(l,h)} = (\boldsymbol{W}^{(l)}\boldsymbol{W}^{(l)\top})^{(h)}, \\
&3)\ \boldsymbol{V}^{(l,h)} = -\boldsymbol{X}^{(l-1)}\boldsymbol{A}^{(l,h)}, \\
&4)\ \boldsymbol{R}^{(l,h)} = \boldsymbol{R}^{(l-1,h)} + \frac{\boldsymbol{Q}^{(l,h)}(\boldsymbol{K}^{(l,h)})^\top}{\sqrt{D_a}}, \\
&5)\ \boldsymbol{T}^{(l,h)} = \kappa^M(\boldsymbol{R}^{(l,h)})\boldsymbol{V}^{(l,h)},
\end{aligned} \tag{4}
$$

where $\boldsymbol{W}^{(l)} = [\boldsymbol{W}^{(l,1)}, ..., \boldsymbol{W}^{(l,H)}] \in \mathbb{R}^{D \times D}$ is a learnable matrix and the submatrix $\boldsymbol{W}^{(l,h)} \in \mathbb{R}^{D \times D_a}$ corresponds to the $h$-th attention head, with $D_a = D/H$. Meanwhile, $(\boldsymbol{W}^{(l)}\boldsymbol{W}^{(l)\top})^{(h)} \in \mathbb{R}^{D \times D_a}$ means that we first compute $\boldsymbol{W}^{(l)}\boldsymbol{W}^{(l)\top}$, denoted as $\boldsymbol{A}^{(l)} = [\boldsymbol{A}^{(l,1)}, ..., \boldsymbol{A}^{(l,H)}]$, where $\boldsymbol{A}^{(l,h)} \in \mathbb{R}^{D \times D_a}$ for $h = 1, ..., H$, and then preserving the $h$-th submatrix $\boldsymbol{A}^{(l,h)}$. Such an operation integrates the functionality of feature-level fusion into the attention heads, where the value matrix of each head involves the model parameters shared across different heads. In addition, $\kappa^M(\cdot)$ denotes the Sinkhorn-scaling algorithm (Cuturi, 2013), passing the attention matrix that involves relational information through an elementwise exponential operation and then normalizing its rows and columns alternatively in $M$ steps, i.e., $\forall \boldsymbol{R} \in \mathbb{R}^{N \times N}$,

$$
\kappa^M(\boldsymbol{R}) = \underbrace{N_c \circ N_r \circ \cdots \circ N_c \circ N_r}_{M \text{ steps}}(\exp(\boldsymbol{R})), \tag{5}
$$

where $N_r(\boldsymbol{R}) = \boldsymbol{R} \oslash (\boldsymbol{R}\mathbf{1}_{N \times N})$ and $N_c(\boldsymbol{R}) = \boldsymbol{R} \oslash (\mathbf{1}_{N \times N}\boldsymbol{R})$ denote row and column normalization operations, and $\oslash$ denotes element-wise division of matrix. Obviously, $\kappa^0$ means the exponential function, i.e., $\kappa^0(\boldsymbol{R}) = \exp(\boldsymbol{R})$.

Finally, given $\{\boldsymbol{T}^{(l,h)}, \boldsymbol{R}^{(l,h)}\}_{h=1}^H$, WGFormer further concatenates them together to derive $\boldsymbol{X}^{(l)}$ and $\boldsymbol{R}^{(l)}$ as

$$
\begin{aligned}
\boldsymbol{X}^{(l)} &= \boldsymbol{X}^{(l-1)} + \text{Concat}(\{\boldsymbol{T}^{(l,h)}\}_{h=1}^H), \\
\boldsymbol{R}^{(l)} &= \text{Concat}(\{\boldsymbol{R}^{(l,h)}\}_{h=1}^H),
\end{aligned} \tag{6}
$$

where $\text{Concat}(\cdot)$ denotes the concatenation operation. Stacking $L$ WGFormer modules together and passing $\{\boldsymbol{X}^{(0)}, \boldsymbol{R}^{(0)}\}$ through them, we derive the final representations $\{\boldsymbol{X}^{(L)}, \boldsymbol{R}^{(L)}\}$, as expressed in Eq. (2).

## 4. Revisiting WGFormer through the Lens of Wasserstein Gradient Flows

### 4.1. Continuous Counterpart of Attention Head

Let's consider a WGFormer with a single attention head (i.e., $H = 1$ and the superscripts $l$ and $h$ in Eq. (4) can be omitted), whose parameter matrix is $\boldsymbol{W} \in \mathbb{R}^{D \times D}$. Given a conformation with $N$ atom-level representations, we introduce a mixture model of the atoms, i.e.,

$$
\mu \in \mathcal{M}(\mathbb{R}^D), \text{ and } \mu = \sum_{i=1}^N \mu_i, \tag{7}
$$

where $\mathcal{M}(\mathbb{R}^D)$ is the space of the measures defined in $\mathbb{R}^D$, and $\mu_i \in \mathcal{M}(\mathbb{R}^D)$ is the measure associated with the $i$-th atom. For an arbitrary measure pair $(\mu_i, \mu_j)$, we describe the relation between $\mu_i$ and $\mu_j$ quantitatively as

$$
r_{ij} = \bar{\boldsymbol{x}}_i^\top \boldsymbol{W}\boldsymbol{W}^\top \bar{\boldsymbol{x}}_j = \bar{\boldsymbol{x}}_i^\top \boldsymbol{A} \bar{\boldsymbol{x}}_j, \tag{8}
$$

where $\bar{\boldsymbol{x}}_i = \int \boldsymbol{x}\mathrm{d}\mu_i(\boldsymbol{x})$ for $i = 1, ..., N$. The collection of the relations leads to a symmetric matrix $\boldsymbol{R} = [r_{ij}]$.

**When each $\mu_i$ is a Dirac measure,** i.e., $\mu_i = \delta_{\boldsymbol{x}_i}$, we can equivalently represent the relations by a function:

$$
r(\boldsymbol{x}, \boldsymbol{x}') = \sum_{i,j=1}^N r_{ij}\delta_{\boldsymbol{x}_i, \boldsymbol{x}_j}(\boldsymbol{x}, \boldsymbol{x}'), \forall \boldsymbol{x}, \boldsymbol{x}' \in \mathbb{R}^D. \tag{9}
$$

In such a situation, the update of the atom-level representations in Eq. (6) is the ResNet equation (He et al., 2016), i.e., $\boldsymbol{X} \to \boldsymbol{X} + \boldsymbol{T}$, which works as a discrete Euler scheme of the ordinary differential equation (ODE) (Weinan, 2017; Satorras et al., 2021; Sander et al., 2022), i.e.,

$$
\frac{\mathrm{d}\boldsymbol{x}_i(t)}{\mathrm{d}t} = T_\mu^M(\boldsymbol{x}_i(t)), \forall i = 1, ..., N. \tag{10}
$$

Here, $\boldsymbol{x}_i(t)$ denotes the $i$-th atom's representation at time $t$, and $T_\mu^M(\boldsymbol{x})$ is an operator for the measure $\mu = \sum_{i=1}^N \mu_i = \sum_{i=1}^N \delta_{\boldsymbol{x}_i}$, which is defined as

$$
T_\mu^M(\boldsymbol{x}) = -\int_{\mathbb{R}^D} k^M(\boldsymbol{x}, \boldsymbol{x}')\boldsymbol{A}\boldsymbol{x}'\mathrm{d}\mu(\boldsymbol{x}'), \tag{11}
$$

where $k^M(\boldsymbol{x}, \boldsymbol{x}')$ is the continuous counterpart of the Sinkhorn module $\kappa^M$ in Eq. (4), i.e.,

$$
\begin{aligned}
&k^0(\boldsymbol{x}, \boldsymbol{x}') = \exp(\boldsymbol{x}^\top \boldsymbol{A}\boldsymbol{x}' + r(\boldsymbol{x}, \boldsymbol{x}')), \\
&k^{m+1}(\boldsymbol{x}, \boldsymbol{x}') = \begin{cases} \frac{k^m(\boldsymbol{x}, \boldsymbol{x}')}{\int k^m(\boldsymbol{x}, \boldsymbol{x}')\mathrm{d}\mu(\boldsymbol{x}')}, & m \text{ is even,} \\ \frac{k^m(\boldsymbol{x}, \boldsymbol{x}')}{\int k^m(\boldsymbol{x}, \boldsymbol{x}')\mathrm{d}\mu(\boldsymbol{x})}, & m \text{ is odd.} \end{cases}
\end{aligned} \tag{12}
$$

Obviously, the collection of $\{T_\mu^M(\boldsymbol{x}_i)\}_{i=1}^n$ is equivalent to the matrix $\boldsymbol{T}$ derived by Eq. (4).

## 4.2. Wasserstein Gradient Flows When $M = 0$ or $\to \infty$

The ODE in Eq. (10) can be equivalently written as a continuity equation (Renardy & Rogers, 2006), i.e.,

$$\partial_t \mu + \mathrm{div}(\mu T_\mu^M) = 0. \tag{13}$$

It has been known that $T_\mu^M$ can be the Wasserstein gradient of an energy function $E : \mathcal{M}(\mathbb{R}^D) \mapsto \mathbb{R}$ at $\mu$, i.e.,

$$T_\mu^M = -\nabla_W E(\mu) := -\nabla\Big(\frac{\delta E}{\delta \mu}(\mu)\Big), \tag{14}$$

where $\frac{\delta E}{\delta \mu}(\mu)$ denotes the first variation of $E$ at $\mu$, which is a function satisfying $\frac{\mathrm{d}E(\mu+\varepsilon\rho)}{\mathrm{d}\varepsilon}|_{\varepsilon=0} = \int \frac{\delta E}{\delta \mu}(\mu)\mathrm{d}\rho$ for every perturbation $\rho$ (Santambrogio, 2017). In this situation, the continuity equation in Eq. (13) captures the evolution of the measure $\mu$ that minimizes the energy function $E$, and each $\boldsymbol{x}(t)$ follows $\frac{\mathrm{d}\boldsymbol{x}(t)}{\mathrm{d}t} = -\nabla_W E(\mu)$ (Jordan et al., 1998).

Based on Proposition 2 in (Sander et al., 2022), we demonstrate that for $\mu = \sum_{i=1}^N \delta_{\boldsymbol{x}_i}$, WGFormer corresponds to Wasserstein gradient flows when $M = 0$ or $M \to \infty$.

**Proposition 4.1.** *For $\mu = \sum_{i=1}^N \delta_{\boldsymbol{x}_i}$, the $T_\mu^M$ in Eq. (11) is the Wasserstein gradient flow, i.e., $T_\mu^M = -\nabla_W E^M(\mu)$, when $M = 0$ or $M \to \infty$. The corresponding energy functions are*

$$E^0(\mu) = \frac{1}{2} \int k^0(\boldsymbol{x}, \boldsymbol{x}')\mathrm{d}\mu(\boldsymbol{x})\mathrm{d}\mu(\boldsymbol{x}'), \tag{15}$$

$$\begin{aligned} E^\infty(\mu) &= \frac{1}{2} \min_{\pi \in \Pi_\mu} \int \log k^0(\boldsymbol{x}, \boldsymbol{x}')\mathrm{d}\pi(\boldsymbol{x}, \boldsymbol{x}') - H(\pi) \\ &= \frac{1}{2} \int k^\infty(\boldsymbol{x}, \boldsymbol{x}') \log \frac{k^0(\boldsymbol{x}, \boldsymbol{x}')}{k^\infty(\boldsymbol{x}, \boldsymbol{x}')} \mathrm{d}\mu(\boldsymbol{x})\mathrm{d}\mu(\boldsymbol{x}'), \end{aligned} \tag{16}$$

*where $\Pi_\mu = \{\pi \in \mathcal{M}(\mathbb{R}^D \times \mathbb{R}^D) | \int_{\boldsymbol{x}} \mathrm{d}\pi(\boldsymbol{x}, \boldsymbol{x}') = \mu(\boldsymbol{x}'), \int_{\boldsymbol{x}'} \mathrm{d}\pi(\boldsymbol{x}, \boldsymbol{x}') = \mu(\boldsymbol{x})\}$, and $H(\pi) = -\int \log(\pi)\mathrm{d}\pi$ is the entropy of $\pi$.*

**Note that, $E^\infty(\mu)$ is an interpretable energy function for conformation optimization.** As shown in Eq. (16), it corresponds to an entropic optimal transport problem (Cuturi, 2013; Peyré et al., 2019). Because $\mu = \sum_{i=1}^N \delta_{\boldsymbol{x}_i}$, we can rewrite $E^\infty(\mu)$ in the following discrete format:

$$\begin{aligned} E^\infty(\mu) &= \min_{\boldsymbol{P} \in \Pi_1} \langle \boldsymbol{X}\boldsymbol{A}\boldsymbol{X}^\top + \boldsymbol{R}, \boldsymbol{P} \rangle + \langle \boldsymbol{P}, \log \boldsymbol{P} \rangle \\ &\Leftrightarrow \max_{\boldsymbol{P} \in \Pi_1} \underbrace{\langle \boldsymbol{D_X}, \boldsymbol{P} \rangle}_{\text{Atomic distance}} \underbrace{- \langle \boldsymbol{P}, \log \boldsymbol{P} \rangle}_{\text{Entropy}}, \end{aligned} \tag{17}$$

where $\boldsymbol{P} \in \Pi_1$ means $\boldsymbol{P}\boldsymbol{1}_N = \boldsymbol{P}^\top\boldsymbol{1}_N = \boldsymbol{1}_N$, and $\boldsymbol{P}$ is the coupling controlling the pairwise relations among the atoms in the latent space. In particular, the first term in Eq. (17) penalizes the cost associated with the interatomic distances

determined by $(\boldsymbol{X}\boldsymbol{A}\boldsymbol{X}^\top + \boldsymbol{R})$. As shown in Eq. (17), this term is equivalent to maximize the expected atomic distance, where the matrix $\boldsymbol{D_X} = [\frac{1}{2}\|\boldsymbol{W}^\top\boldsymbol{x}_i - \boldsymbol{W}^\top\boldsymbol{x}_j\|_2^2 - r_{ij}] \in \mathbb{R}^{N \times N}$ captures the atomic distances in the latent space. In other words, minimizing $E^\infty$ can prevent atoms from being concentrated together in the latent space. On the other hand, the second term in Eq. (17) regularizes the entropy of $\boldsymbol{P}$, which helps build dense and coherent relations among the atoms in the latent space. In our opinion, minimizing such an energy function involving the above two terms is reasonable for conformation optimization for the following two reasons. Firstly, the atoms of the ground-state conformation should have reasonable interatomic distances, avoiding the high energy caused by too close distances. Secondly, the energy of conformation is calculated based on dense atomic pairs rather than sparse chemical bond-based connections.

In summary, by stacking $L$ WGFormer layers and setting a large value for the hyperparameter $M$, our model achieves the Wasserstein gradient flows defined on the measure of atoms. Given the initial measure $\mu^{(0)}$ corresponding to $\boldsymbol{X}^{(0)}$ and the initial relational information $\boldsymbol{R}^{(0)}$, each layer updates the measure, i.e., $\mu^{(l+1)} = \arg\min_\mu E^\infty(\mu) + W_2(\mu, \mu^{(l+1)})$, where $W_2(\cdot, \cdot)$ denotes the 2-order Wasserstein distance, and update the relational information $\boldsymbol{R} = [r_{ij}]$ based on the new measure (by Eq. (8)).

**When each $\mu_i$ is a non-Dirac measure,** our WGFormer can still achieve Wasserstein gradient flows when $M = 0$. For each non-Dirac $\mu_i$, we can define an operator as

$$T_{\mu_i}^0(\boldsymbol{x}) = -\sum_{j=1}^N \int \exp(\boldsymbol{x}^\top\boldsymbol{A}\boldsymbol{x}' + r_{ij})\boldsymbol{A}\boldsymbol{x}'\mathrm{d}\mu_j(\boldsymbol{x}'). \tag{18}$$

Then, we have the following theorem:

**Proposition 4.2.** *Let $E : \mathcal{M}(\mathbb{R}^D)^N \mapsto \mathbb{R}$ be an energy function for $\{\mu_i\}_{i=1}^N$, which is defined as*

$$E = \frac{1}{2}\sum_{i,j=1}^N \int \exp(\boldsymbol{x}^\top\boldsymbol{A}\boldsymbol{x}' + r_{ij})\mathrm{d}\mu_i(\boldsymbol{x})\mathrm{d}\mu_j(\boldsymbol{x}'). \tag{19}$$

*For $i = 1, ..., N$, $T_{\mu_i}^0$ is the Wasserstein gradient flow that $\min_{\mu_i} E$, i.e., $T_{\mu_i}^0 = -\nabla_W E(\mu_i; \{\mu_j\}_{j \neq i})$.*

However, for non-Dirac $\mu_i$'s, WGFormer does not correspond to Wasserstein gradient flows in general when $M \to \infty$ because in this case the relational information $r_{ij}$ in Eq. (8) cannot be rewritten as the function in Eq. (9). The proofs of the above propositions are shown in Appendix B.

## 5. Learning Algorithm

Given the predicted ground-state conformation $\widehat{\boldsymbol{C}}$, we can derive the corresponding interatomic distances $\widehat{\boldsymbol{D}}$. Existing learning-based methods merely consider penalizing the

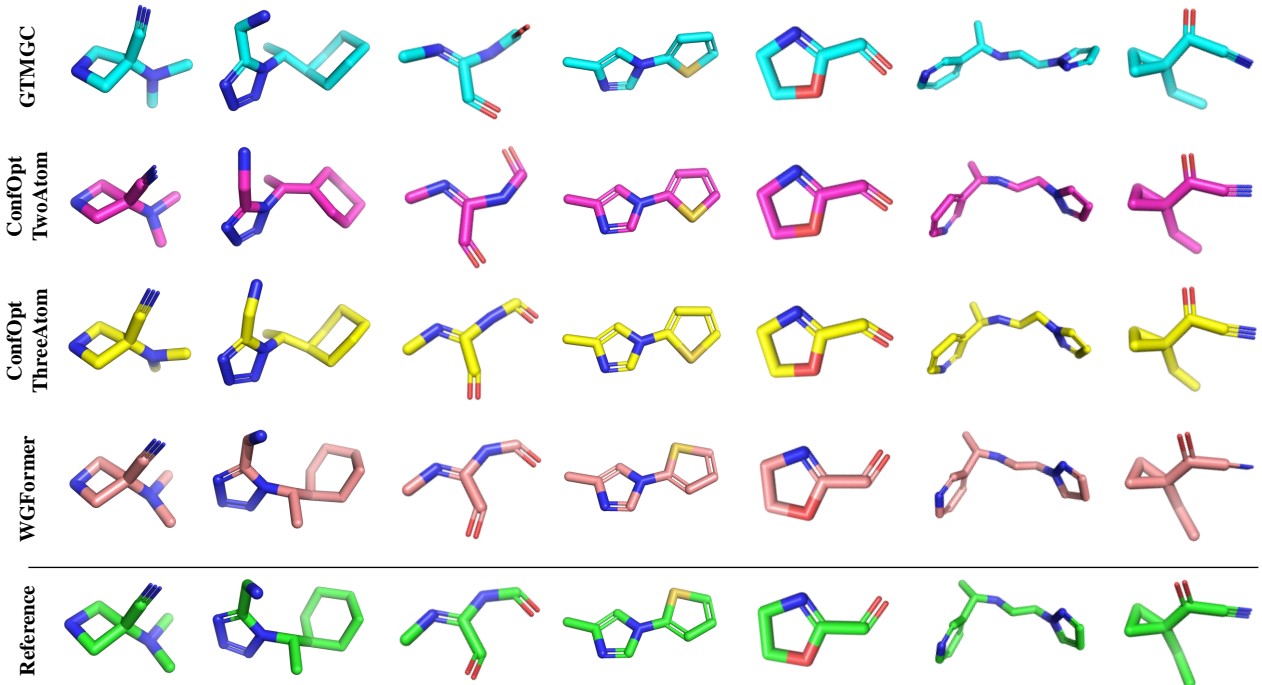

*Figure 3.* The visual comparison for the ground-state conformations predicted by our WGFormer and the top-3 baselines.

mean-absolute-error (MAE) between $\widehat{D}$ and the ground truth $D$, which may not provide sufficient supervision and thus limits the model performance (Guan et al., 2022; Xu et al., 2023; Kim et al., 2025). In this work, we design a multi-task learning strategy to train the proposed model, making the model fit the ground-state conformation $C$ and the corresponding interatomic distances $D$ jointly. In particular, given $\widehat{C}$ and $C$, we firstly align $C$ to $\widehat{C}$ i.e.,

$$C_0 = C - \frac{1}{N} \sum_{i=1}^{N} c_i, \ \ \widehat{C}_0 = \widehat{C} - \frac{1}{N} \sum_{i=1}^{N} \hat{c}_i, \quad (20)$$
$$C^* = C_0 \mathrm{Rot}(C_0, \widehat{C}_0) = C_0 V U^\top,$$

where $C_0$ and $\widehat{C}_0$ are zero-mean conformations, and $C^*$ is the aligned ground truth. $\mathrm{Rot}(C_0, \widehat{C}_0)$ is the rotation matrix derived by Procrustes analysis (Wang & Mahadevan, 2008), in which $U$ and $V$ are obtained from the singular value decomposition (SVD) of $\widehat{C}_0^\top C_0$.

The final loss function is defined as follows:

$$\mathcal{L} = \underbrace{\frac{1}{N^2} \|\widehat{D} - D\|_1}_{\mathrm{MAE}(\widehat{D}, D)} + \lambda \underbrace{\sqrt{\frac{1}{N} \|\widehat{C}_0 - C^*\|_F^2}}_{\mathrm{RMSD}(\widehat{C}_0, C^*)}. \quad (21)$$

where $\mathrm{MAE}(\widehat{D}, D)$ denotes the mean-absolute-error between $\widehat{D}$ and the ground truth $D$, which is the main loss supervising the corresponding interatomic distances. $\mathrm{RMSD}(\widehat{C}_0, C^*)$ denotes the root-mean-square-deviation between $\widehat{C}_0$ and the ground truth $C^*$, which is the auxiliary

loss supervising the predicted ground-state conformation. $\lambda \geq 0$ controls the significance of the auxiliary loss.

## 6. Experiments

To demonstrate the effectiveness of our method, we compare it with state-of-the-art methods. Meanwhile, we also conduct comprehensive ablation studies and analyses to validate its rationality and superiority. Experimental details and results are provided in this section and Appendix C.

### 6.1. Experimental Setup

**Datasets.** Following previous work (Xu et al., 2023), we employ the widely used Molecule3D and QM9 datasets to validate our method. In particular, Molecule3D (Xu et al., 2021d) is a large-scale dataset, comprising approximately four million molecules, and provides two dataset-splitting results (i.e., random-splitting and scaffold-splitting) to ensure a comprehensive evaluation. QM9 (Ramakrishnan et al., 2014) is a small-scale quantum chemistry dataset, comprising approximately 130,000 molecules with up to 9 heavy atoms, and adopts the identical dataset-splitting strategy described in (Liao & Smidt, 2023). Each molecule in these datasets corresponds to the ground-state conformation determined through density functional theory (DFT) calculations.

**Evaluation.** To guarantee a comprehensive and fair evaluation, we employ the same metrics used in (Xu et al., 2023), including mean-absolute-error of distances (D-MAE), root-

*Table 1.* The performance comparison for various methods. For each dataset, we bold the best result and underline the second-best one.

| Dataset | Method | Model | Validation | | | Test | | |
|---|---|---|---|---|---|---|---|---|
| | | | D-MAE ↓ | D-RMSE ↓ | C-RMSD ↓ | D-MAE ↓ | D-RMSE ↓ | C-RMSD ↓ |
| Molecule3D (random) | 2D Method | GINE | 0.590 | 1.014 | 1.116 | 0.592 | 1.018 | 1.116 |
| | | GATv2 | 0.563 | 0.983 | 1.082 | 0.564 | 0.986 | 1.083 |
| | | GPS | 0.528 | 0.909 | 1.036 | 0.529 | 0.911 | 1.038 |
| | | GTMGC | 0.432 | 0.719 | 0.712 | 0.433 | 0.721 | 0.713 |
| | 3D Method | SE(3)-Transformer | 0.466 | 0.712 | 0.800 | 0.467 | 0.774 | 0.802 |
| | | EGNN | 0.461 | 0.704 | 0.798 | 0.462 | 0.766 | 0.799 |
| | | ConfOpt-TwoAtom | 0.438 | 0.668 | 0.748 | 0.438 | 0.670 | 0.749 |
| | | ConfOpt-ThreeAtom | 0.429 | 0.659 | 0.734 | 0.430 | 0.661 | 0.736 |
| | | WGFormer (ours) | **0.391** | **0.649** | **0.662** | **0.392** | **0.652** | **0.664** |
| Molecule3D (scaffold) | 2D Method | GINE | 0.883 | 1.517 | 1.407 | 1.400 | 2.224 | 1.960 |
| | | GATv2 | 0.778 | 1.385 | 1.254 | 1.238 | 2.069 | 1.752 |
| | | GPS | 0.538 | 0.885 | 1.031 | 0.657 | 1.091 | 1.136 |
| | | GTMGC | 0.406 | 0.675 | 0.678 | 0.400 | 0.679 | 0.693 |
| | 3D Method | SE(3)-Transformer | 0.460 | 0.676 | 0.775 | 0.456 | 0.678 | 0.747 |
| | | EGNN | 0.448 | 0.666 | 0.758 | 0.442 | 0.670 | 0.741 |
| | | ConfOpt-TwoAtom | 0.408 | 0.626 | 0.708 | 0.402 | 0.628 | 0.698 |
| | | ConfOpt-ThreeAtom | 0.401 | 0.619 | 0.697 | 0.395 | 0.622 | 0.691 |
| | | WGFormer (ours) | **0.363** | **0.599** | **0.618** | **0.360** | **0.610** | **0.627** |
| QM9 | 2D Method | GINE | 0.357 | 0.673 | 0.685 | 0.357 | 0.669 | 0.693 |
| | | GATv2 | 0.339 | 0.663 | 0.661 | 0.339 | 0.659 | 0.666 |
| | | GPS | 0.326 | 0.644 | 0.662 | 0.326 | 0.640 | 0.666 |
| | | GTMGC | 0.262 | 0.468 | 0.362 | 0.264 | 0.470 | 0.367 |
| | 3D Method | SE(3)-Transformer | 0.254 | 0.451 | 0.296 | 0.256 | 0.455 | 0.303 |
| | | EGNN | 0.248 | 0.442 | 0.257 | 0.251 | 0.449 | 0.265 |
| | | ConfOpt-TwoAtom | 0.245 | 0.439 | 0.245 | 0.248 | 0.444 | 0.254 |
| | | ConfOpt-ThreeAtom | 0.241 | 0.433 | 0.237 | 0.244 | 0.438 | 0.246 |
| | | WGFormer (ours) | **0.223** | **0.416** | **0.198** | **0.227** | **0.422** | **0.206** |

mean-squared-error of distances (D-RMSE), and root-mean-square-deviation of coordinates (C-RMSD), to evaluate the performance of our method. The specific definitions of these evaluation metrics are expressed as follows:

$$\text{D-MAE}(\widehat{\mathcal{D}}, \mathcal{D}^*) = \frac{1}{N^2} \sum_{i=1}^{N} \sum_{j=1}^{N} |\hat{d}_{ij} - d_{ij}^*|, \quad (22)$$

$$\text{D-RMSE}(\widehat{\mathcal{D}}, \mathcal{D}^*) = \sqrt{\frac{1}{N^2} \sum_{i=1}^{N} \sum_{j=1}^{N} (\hat{d}_{ij} - d_{ij}^*)^2}, \quad (23)$$

$$\text{C-RMSD}(\widehat{\mathcal{C}}, \mathcal{C}^*) = \sqrt{\frac{1}{N} \sum_{i=1}^{N} \|\hat{c}_i - c_i^*\|_2^2}. \quad (24)$$

where $\widehat{\mathcal{C}} = [\hat{c}_i] \in \mathbb{R}^{N \times 3}$ is predicted ground-state conformation (zero-mean), $\mathcal{C}^* = [c_i^*] \in \mathbb{R}^{N \times 3}$ is the ground-truth aligned through Eq. (20), $N$ is the number of atoms in the molecule, and $\hat{d}_{ij} = \|\hat{c}_i - \hat{c}_j\|_2$, $d_{ij}^* = \|c_i^* - c_j^*\|_2$.

**Baselines.** To demonstrate the effectiveness of our method, state-of-the-art methods in two typical categories are employed as our baselines. The first category is the 2D methods predicting ground-state conformations merely based on 2D molecular graphs, including GTMGC (Xu et al.,

2023) and its various variants that use GINE (Hu et al., 2020), GATv2 (Brody et al., 2021), and GPS (Rampášek et al., 2022) as backbones. The second category is the 3D methods predicting ground-state conformations via conformation optimization, including ConfOpt (Guan et al., 2022) (TwoAtom and ThreeAtom versions) and its various variants that use SE(3)-Transformer (Fuchs et al., 2020) and EGNN (Satorras et al., 2021) as backbones. All these baselines are implemented based on their default settings.

**Model architecture.** Our model comprises 30 WGFormer layers, each equipped with 64 attention heads. The atom-level and relational representation dimensions are set to 512 and 64, respectively. We train the whole model on a single RTX 3090 GPU and select the optimal checkpoint based on performance evaluated on the corresponding validation set.

### 6.2. Comparisons

Table 1 presents the performance of various methods, showing that our WGFormer consistently outperforms all baselines across all evaluation metrics on all datasets. Specifically, compared to the best baseline, our WGFormer reduces C-RMSD (the most critical metric in this task) by nearly 10% on Molecule3D and 15% on QM9, demonstrating its

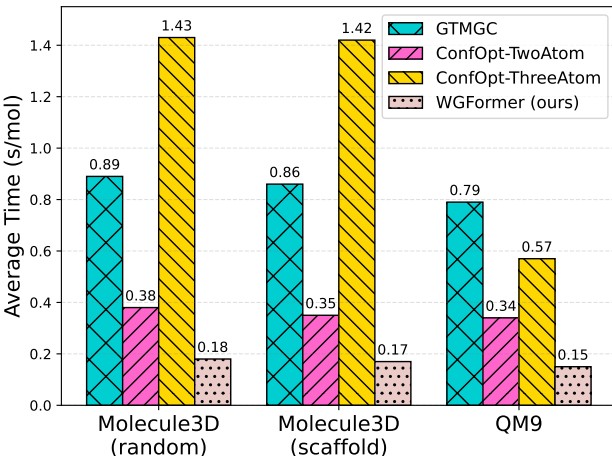

*Figure 4.* The efficiency comparison for our WGFormer and the top-3 baselines on each dataset.

*Table 2.* The ablation study of multi-task learning strategy.

| | $\mathcal{L}_{\text{MAE}}$ | $\mathcal{L}_{\text{RMSD}}$ | D-MAE ↓ | D-RMSE ↓ | C-RMSD ↓ |
|---|---|---|---|---|---|
| Validation | × | ✓ | 0.652 | 0.921 | 0.716 |
| | ✓ | × | **0.384** | 0.657 | 0.688 |
| | ✓ | ✓ | 0.391 | **0.649** | **0.662** |
| Test | × | ✓ | 0.653 | 0.924 | 0.718 |
| | ✓ | × | **0.386** | 0.660 | 0.689 |
| | ✓ | ✓ | 0.392 | **0.652** | **0.664** |

*Table 3.* The ablation study of our attention head.

| | Sinkhorn Module | QKV Adjust | D-MAE ↓ | D-RMSE ↓ | C-RMSD ↓ |
|---|---|---|---|---|---|
| Validation | × | × | 0.379 | 0.628 | 0.661 |
| | ✓ | × | 0.378 | 0.631 | 0.656 |
| | × | ✓ | 0.379 | 0.625 | 0.658 |
| | ✓ | ✓ | **0.363** | **0.599** | **0.618** |
| Test | × | × | 0.375 | 0.636 | 0.662 |
| | ✓ | × | 0.376 | 0.644 | 0.662 |
| | × | ✓ | 0.376 | 0.635 | 0.665 |
| | ✓ | ✓ | **0.360** | **0.610** | **0.627** |

exceptional capability. Meanwhile, our WGFormer also achieves consistent dominance across both random and scaffold splits of Molecule3D, demonstrating its effectiveness under different data-splitting strategies. In addition, we have visualized some ground-state conformations predicted by our WGFormer and the top-3 baselines (GTMGC, ConfOpt-TwoAtom, and ConfOpt-ThreeAtom) in Figure 3. As shown in this figure, the ground-state conformations predicted by our WGFormer align more closely with the ground truth, further demonstrating its superiority.

Figure 4 compares the efficiency of our WGFormer and the top-3 baselines by calculating the average time taken per molecule on each dataset. Even if our model applies 30 WGFormer layers, it is significantly faster than baselines, reducing the runtime per molecule by over 50%. One important reason for this superior performance is the simplicity of our WGFomer architecture, which can update the atom-level and relational representations simultaneously. On the contrary, each layer of ConfOpt applies two modules to update the atom-level and relational representations sequentially, and ConfOpt-ThreeAtom further considers the relations within the triplets of atoms, significantly increasing the model complexity and computational cost. In summary, the above quantitative and qualitative comparisons highlight the superiority of our WGFormer in both accuracy and efficiency for ground-state conformation prediction.

### 6.3. Ablation Studies and Analyses

**Rationality of multi-task learning.** We conduct an ablation study of our multi-task learning strategy on random-split Molecule3D. As shown in Table 2, merely minimizing RMSD($\widehat{C}_0, C^*$) performs significantly worse than minimizing MAE($\widehat{D}, D$). This phenomenon aligns with previous works prioritizing optimizing interatomic distances

over atomic 3D coordinates, proving the importance of emphasizing optimizing interatomic distances during training. Moreover, compared with merely minimizing a single loss, minimizing them jointly achieves better performance, especially in C-RMSD (the most critical metric), thus supporting the rationality of our multi-task learning strategy.

**Rationality of WGFormer architecture.** In addition to our theoretical analysis in Section 4, we also validate the rationality of our model architecture through experiments. Here, we conduct an ablation study for the proposed Sinkhorn-based attention head on scaffold-split Molecule3D. As described in Section 3.2, our WGFormer applies the Sinkhorn-scaling algorithm with the proposed "QKV" adjustment. The results in Table 3 show that $i$) each single modification does not harm performance, and $ii$) applying the two modifications jointly leads to significant performance improvements, especially in C-RMSD (the most critical metric). These empirical findings align closely with our theoretical analysis in Section 4, further validating the rationality behind the proposed architecture design.

**Correlative analysis of latent energy minimization and conformation optimization.** As analyzed theoretically in Section 4, our WGFormer optimizes conformations by minimizing an energy function (i.e., Eq. (17)) defined on the latent mixture models of atoms. To validate that minimizing this latent energy correlates with conformation optimization, we conduct a series of analytic experiments on QM9. In particular, given a trained WGFormer with 30 layers, we first randomly sample some molecules from the test set and pass them through the trained WGFormer.

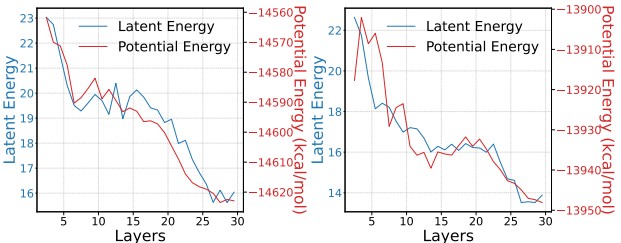

*Figure 5.* Some cases about the change of latent and potential energy as the number of layers increases, where latent energy is obtained by solving Eq. (17) and potential energy is obtained by xTB (Bannwarth et al., 2019). More cases are provided in Figure 9.

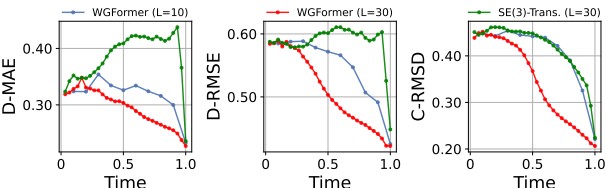

*Figure 6.* The comparison for WGFormer$_{10}$, WGFormer$_{30}$ and SE(3)-Transformer$_{30}$ on their optimization processes from $t = 0$ to $t = 1$. For each model and $l = 1, ..., L$, the point at $t = \frac{l}{L}$ corresponds to the model's performance achieved by $l$ layers.

*Table 4.* The mean energy change across layers.

| Layers | 5 | 10 | 15 | 20 | 25 | 30 |
|---|---|---|---|---|---|---|
| Latent energy | -3.629 | -7.729 | -8.512 | -8.932 | -9.195 | -10.385 |
| Potential energy | -9.135 | -18.199 | -19.955 | -34.814 | -45.204 | -52.378 |

Then, based on the outputs (i.e., $X$ and $R$) of each layer, we can calculate the latent energy per layer by solving Eq. (17). As illustrated in Figure 5, the latent energy decreases gradually as the number of layers increases, which validates that our WGFormer is indeed minimizing this latent energy.

Furthermore, considering that conformation optimization corresponds to minimizing the potential energy as the ground-state conformation is the energy-minimized conformation, we employ the widely used xTB tool (Bannwarth et al., 2019) to calculate the corresponding potential energy of conformations obtained by different layers (i.e., passing the outputs of each layer through the trained decoder). As shown in Figure 5, the potential energy indeed decreases gradually as the number of layers increases. Taking the latent and potential energy of the first layer as references, we calculate their relative change across increasing layers (the mean values are reported in Table 4) to analyze the correlation between the latent and potential energy. Here, the high Pearson correlation coefficient ($0.885 \pm 0.033$) indicates a strong linear relationship, and the marginally higher distance correlation ($0.906 \pm 0.018$) suggests the presence of additional nonlinear dependencies. These results validate that minimizing this latent energy indeed drives potential energy minimization, thereby being highly correlated with conformation optimization.

**Impact analysis of layer numbers on conformation optimization.** As discussed earlier, our model corresponds to minimizing the latent energy function defined in Eq. (17), and each layer works as an update step in the Euler scheme of ODE. Specifically, without the loss of generality, we assume that the optimization progress starts at $t = 0$ and ends at $t = 1$. In this case, learning our model with $L$ WGFormer layers results in applying $L$ optimization steps in the time

interval $[0, 1]$, and accordingly, the step size is $\Delta t = \frac{1}{L}$.

We quantitatively analyze the impact of layer numbers and thus further verify the optimization nature of our model. In particular, we train two models with 10 and 30 WGFormer layers on QM9, respectively. Given each trained model, we evaluate its performance using different layer numbers on the test set, so that we can observe the optimization process achieved by each model to analyze the convergence rate and optimization efficiency. As illustrated in Figure 6, the performance of our models improves steadily as the number of layers increases. Meanwhile, the 30-layer WGFormer model performs better than the 10-layer WGFormer model, which matches the common sense of optimization: Learning a large number of layers means applying fine-grained optimization with more steps, which leads to lower energy and improved model performance while increasing runtime. Besides, these results further validate that our model indeed achieves conformation optimization.

In addition, as shown in Figure 6, when applying a non-WGFormer architecture (e.g., an original SE(3)-Transformer removing our modifications) and keeping other settings unchanged, its performance may not be improved consistently when increasing the number of layers. This phenomenon further demonstrates the lack of interpretability in existing methods, thereby highlighting the value of our WGFormer.

## 7. Conclusion

In this work, we propose WGFormer, a Wasserstein gradient flow-driven SE(3)-Transformer that can effectively tackle the molecular ground-state conformation prediction task. In particular, our WGFormer can be interpreted as Wasserstein gradient flows, which optimizes molecular conformation by minimizing a physically reasonable energy function defined on the latent mixture models of atoms, thereby significantly improving performance and interpretability. In the future, we will continue to explore the applications of our method in various conformation-related tasks. Besides, we will further study the mathematics of our WGFormer to design more powerful and interpretable model architectures.

## Acknowledgements

This work was supported by the National Natural Science Foundation of China (92270110), the Fundamental Research Funds for the Central Universities, the Research Funds of Renmin University of China, and the Public Computing Cloud, Renmin University of China. We also acknowledge the support provided by the fund for building world-class universities (disciplines) of Renmin University of China and by the funds from Beijing Key Laboratory of Research on Large Models and Intelligent Governance, Engineering Research Center of Next-Generation Intelligent Search and Recommendation, Ministry of Education, and from Intelligent Social Governance Interdisciplinary Platform, Major Innovation & Planning Interdisciplinary Platform for the "Double-First Class" Initiative, Renmin University of China.

## Impact Statement

This paper presents work that aims to advance the field of Machine Learning and AI for Science, especially for molecular modeling and ground-state conformation prediction. There are many potential societal consequences of our work, none of which we feel must be specifically highlighted here.

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

# A. SE(3)-Equivariance

To prove our whole auto-encoding framework is SE(3)-equivariant, we will verify that if the input low-quality conformation $\widetilde{\mathcal{C}} = [\tilde{c}_i] \in \mathbb{R}^{N \times 3}$ undergo an SE(3)-transformation $(\mathbf{Q}, \mathbf{t})$, where $\mathbf{Q} \in \mathbb{R}^{3 \times 3}$ is a rotation matrix and $\mathbf{t} \in \mathbb{R}^3$ is a translation, the predicted ground-state conformation $\widehat{\mathcal{C}} = [\hat{c}_i] \in \mathbb{R}^{N \times 3}$ will also transform in the same manner. Specifically, let input $\widetilde{\mathcal{C}}' = [\tilde{c}_i'] = [\mathbf{Q}\tilde{c}_i + \mathbf{t}]$, we aim to demonstrate that the corresponding output satisfies $\widehat{\mathcal{C}}' = [\hat{c}_i'] = [\mathbf{Q}\hat{c}_i + \mathbf{t}]$.

In the **encoding phase**, the initial atom-level representation $\boldsymbol{X}^{(0)} = [\boldsymbol{x}_i^{(0)}] \in \mathbb{R}^{N \times D}$ can be extracted as follows:

$$\boldsymbol{x}_i^{(0)} = f(v_i), \tag{25}$$

where $f$ is an embedding function dependent only on the atom type $v_i$. Since atom type is independent of the geometric coordinates, the initial atom-level representation $\boldsymbol{X}^{(0)} = [\boldsymbol{x}_i^{(0)}]$ remains unchanged under the SE(3)-transformation.

Meanwhile, the initial interatomic relational representation $\boldsymbol{R}^{(0)} = [r_{ij}^{(0)}] \in \mathbb{R}^{N \times N \times H}$ can be extracted as follows:

$$\boldsymbol{r}_{ij}^{(0)} = \mathcal{N}(\tilde{d}_{ij}\boldsymbol{u}_{v_i v_j} + \boldsymbol{v}_{v_i v_j}; \boldsymbol{\mu}, \boldsymbol{\sigma}), \tag{26}$$

where $\tilde{d}_{ij} = \|\tilde{c}_i - \tilde{c}_j\|_2$ is the interatomic distance, which remains unchanged under the SE(3)-transformation $(\mathbf{Q}, \mathbf{t})$, i.e.,

$$\tilde{d}_{ij}' = \|\tilde{c}_i' - \tilde{c}_j'\|_2 = \|(\mathbf{Q}\tilde{c}_i + \mathbf{t}) - (\mathbf{Q}\tilde{c}_j + \mathbf{t})\|_2 = \|\mathbf{Q}\tilde{c}_i - \mathbf{Q}\tilde{c}_j\|_2 = \|\tilde{c}_i - \tilde{c}_j\|_2 = \tilde{d}_{ij}, \tag{27}$$

The initial interatomic relational representation $\boldsymbol{R}^{(0)} = [r_{ij}^{(0)}]$ also remains unchanged under the SE(3)-transformation.

Furthermore, since the initial atom-level representation $\boldsymbol{X}^{(0)} = [\boldsymbol{x}_i^{(0)}]$ and interatomic relational representation $\boldsymbol{R}^{(0)} = [r_{ij}^{(0)}]$ remains unchanged under the SE(3)-transformation, the final atom-level representation $\boldsymbol{X}^{(L)} = [\boldsymbol{x}_i^{(L)}]$ and interatomic relational representation $\boldsymbol{R}^{(L)} = [r_{ij}^{(L)}]$ will also remain unchanged under the SE(3)-transformation. In other words, $\boldsymbol{X}^{(0)} = [\boldsymbol{x}_i^{(0)}]$, $\boldsymbol{R}^{(0)} = [r_{ij}^{(0)}]$, $\boldsymbol{X}^{(L)} = [\boldsymbol{x}_i^{(L)}]$ and $\boldsymbol{R}^{(L)} = [r_{ij}^{(L)}]$ are all SE(3)-invariant.

In the **decoding phase**, the predicted ground-state conformation $\widehat{\mathcal{C}} = [\hat{c}_i]$ can be computed as follows:

$$\hat{c}_i = \tilde{c}_i + \sum\nolimits_{j=1}^N \frac{\text{MLP}(r_{ij}^{(L)} - r_{ij}^{(0)})(\tilde{c}_i - \tilde{c}_j)}{N}, \tag{28}$$

Under the SE(3)-transformation $(\mathbf{Q}, \mathbf{t})$, the input coordinates $\tilde{c}_i'$ and $\tilde{c}_j'$ are transformed as:

$$\tilde{c}_i' = \mathbf{Q}\tilde{c}_i + \mathbf{t}, \quad \tilde{c}_j' = \mathbf{Q}\tilde{c}_j + \mathbf{t}, \tag{29}$$

The relative displacements in the decoding equation are then given by:

$$\tilde{c}_i' - \tilde{c}_j' = (\mathbf{Q}\tilde{c}_i + \mathbf{t}) - (\mathbf{Q}\tilde{c}_j + \mathbf{t}) = \mathbf{Q}(\tilde{c}_i - \tilde{c}_j), \tag{30}$$

As the MLP operates on SE(3)-invariant inputs $(r_{ij}^{(L)} - r_{ij}^{(0)})$, the updated predicted coordinates become:

$$\hat{c}_i' = \tilde{c}_i' + \sum\nolimits_{j=1}^N \frac{\text{MLP}(r_{ij}^{(L)} - r_{ij}^{(0)})(\tilde{c}_i' - \tilde{c}_j')}{N}, \tag{31}$$

Expanding $\tilde{c}_i'$ and $\tilde{c}_j'$ in terms of $\mathbf{Q}$ and $\mathbf{t}$, we have:

$$\hat{c}_i' = (\mathbf{Q}\tilde{c}_i + \mathbf{t}) + \sum\nolimits_{j=1}^N \frac{\text{MLP}(r_{ij}^{(L)} - r_{ij}^{(0)})\mathbf{Q}(\tilde{c}_i' - \tilde{c}_j')}{N}, \tag{32}$$

Since the rotation matrix $\mathbf{Q}$ is a linear transformation, it factors out from the weighted sum:

$$\hat{c}_i' = \mathbf{Q}\left(\tilde{c}_i + \sum\nolimits_{j=1}^N \frac{\text{MLP}(r_{ij}^{(L)} - r_{ij}^{(0)})(\tilde{c}_i - \tilde{c}_j)}{N}\right) + \mathbf{t}, \tag{33}$$

The term in parentheses is simply the original $\hat{c}_i$, i.e.,

$$\hat{c}_i' = \mathbf{Q}\hat{c}_i + \mathbf{t}, \tag{34}$$

Therefore, our whole auto-encoding framework is SE(3)-equivariant.

# B. Proofs of Key Theoretical Results

In principle, the proofs of our theoretical results are based on the techniques used in (Sander et al., 2022). However, because of considering the interatomic relations and their updates, our WGFormer corresponds to Wasserstein gradient flows under a stricter condition, and the energy function it minimizes considers the relational information of molecular conformation.

## B.1. Proof of Proposition 4.1

*Proof.* Recall that $\mu = \sum_{i=1}^N \mu_i$, $\mu_i \in \mathcal{M}(\mathbb{R}^D)$ for $i = 1, ..., N$. In addition, we impose the standard sufficient regularity assumption on each $\mu_i$. When each component measure is a Dirac measure, i.e., $\mu_i = \delta_{\boldsymbol{x}_i}$, we have $\mu = \sum_{i=1}^N \delta_{\boldsymbol{x}_i}$. In such a situation, we rewrite the energy function in Eq. (15) as

$$E^0(\mu) = \frac{1}{2} \iint_{\mathbb{R}^D \times \mathbb{R}^D} \underbrace{\exp(\boldsymbol{x}^\top \boldsymbol{A} \boldsymbol{x}' + r(\boldsymbol{x}, \boldsymbol{x}'))}_{k^0(\boldsymbol{x}, \boldsymbol{x}')} \mathrm{d}\mu(\boldsymbol{x}) \mathrm{d}\mu(\boldsymbol{x}') = \frac{1}{2} \sum_{i,j=1}^N \exp(\boldsymbol{x}_i^\top \boldsymbol{A} \boldsymbol{x}_j + r_{ij}). \tag{35}$$

Accordingly, its first variation at $\mu$ is

$$\frac{\delta E^0}{\delta \mu}(\mu)(\boldsymbol{x}) = \int_{\mathbb{R}^D} \exp(\boldsymbol{x}^\top \boldsymbol{A} \boldsymbol{x}' + r(\boldsymbol{x}, \boldsymbol{x}')) \mathrm{d}\mu(\boldsymbol{x}') = \sum_{j=1}^N \exp(\boldsymbol{x}^\top \boldsymbol{A} \boldsymbol{x}_j + r(\boldsymbol{x}, \boldsymbol{x}_j)). \tag{36}$$

In this case, the operator $T_\mu^0(\boldsymbol{x})$ becomes

$$T_\mu^0(\boldsymbol{x}) = -\int_{\mathbb{R}^D} \exp(\boldsymbol{x}^\top \boldsymbol{A} \boldsymbol{x}' + r(\boldsymbol{x}, \boldsymbol{x}')) \boldsymbol{A} \boldsymbol{x}' \mathrm{d}\mu(\boldsymbol{x}') = -\sum_{j=1}^N \exp(\boldsymbol{x}^\top \boldsymbol{A} \boldsymbol{x}_j + r(\boldsymbol{x}, \boldsymbol{x}_j)) \boldsymbol{A} \boldsymbol{x}_j. \tag{37}$$

Then, we have

$$\nabla_W E^0(\mu)(\boldsymbol{x}) = \sum_{j=1}^N \nabla_{\boldsymbol{x}} \exp(\boldsymbol{x}^\top \boldsymbol{A} \boldsymbol{x}_j + r(\boldsymbol{x}, \boldsymbol{x}_j)) = \sum_{j=1}^N \exp(\boldsymbol{x}^\top \boldsymbol{A} \boldsymbol{x}_j + r(\boldsymbol{x}, \boldsymbol{x}_j)) \boldsymbol{A} \boldsymbol{x}_j = -T_\mu^0(\boldsymbol{x}). \tag{38}$$

In general, given $\mu = \sum_{i=1}^N \mu_i$, we define an energy function of $\{\mu_i\}_{i=1}^N$ with $M \to \infty$ as

$$E^\infty(\{\mu_i\}_{i=1}^N) = \frac{1}{2} \inf_{\pi \in \Pi(\mu,\mu)} \sum_{i,j=1}^N \iint_{\mathbb{R}^D \times \mathbb{R}^D} \pi(\boldsymbol{x}, \boldsymbol{x}')(\boldsymbol{x}^\top \boldsymbol{A} \boldsymbol{x}' + r_{ij}) + \pi(\boldsymbol{x}, \boldsymbol{x}') \log \pi(\boldsymbol{x}, \boldsymbol{x}') \mathrm{d}\mu_i(\boldsymbol{x}) \mathrm{d}\mu_j(\boldsymbol{x}')$$

$$= \frac{1}{2} \inf_{\pi \in \Pi(\mu,\mu)} \sum_{i,j=1}^N \iint_{\mathbb{R}^D \times \mathbb{R}^D} \pi(\boldsymbol{x}, \boldsymbol{x}')(\boldsymbol{x}^\top \boldsymbol{A} \boldsymbol{x}' + r_{ij}) \mathrm{d}\mu_i(\boldsymbol{x}) \mathrm{d}\mu_j(\boldsymbol{x}') - H(\pi), \tag{39}$$

where $H(\pi) = -\iint_{\mathbb{R}^D \times \mathbb{R}^D} \pi(\boldsymbol{x}, \boldsymbol{x}') \mathrm{d}\mu(\boldsymbol{x}) \mathrm{d}\mu(\boldsymbol{x}')$ is the entropy of the coupling $\pi$. Essentially, Eq. (39) corresponds to an entropic optimal transport (EOT) problem defined for the mixture model $\mu$.

Furthermore, when $\mu = \sum_{i=1}^N \delta_{\boldsymbol{x}_i}$, we can equivalently rewrite it as a function of $\mu$ and in a classic EOT format, i.e.,

$$E^\infty(\mu) = \frac{1}{2} \min_{\pi \in \Pi(\mu,\mu)} \iint_{\mathbb{R}^D \times \mathbb{R}^D} \underbrace{(\boldsymbol{x}^\top \boldsymbol{A} \boldsymbol{x}' + r(\boldsymbol{x}, \boldsymbol{x}'))}_{\text{cost: } \log k^0(\boldsymbol{x}, \boldsymbol{x}')} \pi(\boldsymbol{x}, \boldsymbol{x}') \mathrm{d}\mu(\boldsymbol{x}) \mathrm{d}\mu(\boldsymbol{x}') - H(\pi)$$

$$= \frac{1}{2} \int k^\infty(\boldsymbol{x}, \boldsymbol{x}') \log \frac{k^0(\boldsymbol{x}, \boldsymbol{x}')}{k^\infty(\boldsymbol{x}, \boldsymbol{x}')} \mathrm{d}\mu(\boldsymbol{x}) \mathrm{d}\mu(\boldsymbol{x}')$$

$$= \frac{1}{2} \max_{f \in \mathcal{C}(\mathbb{R}^D)} f(\boldsymbol{x}) + f^c(\boldsymbol{x}) \mathrm{d}\mu(\boldsymbol{x}). \tag{40}$$

Here, the second equality in Eq. (40) means applying the Sinkhorn-scaling algorithm to solve the EOT problem (with $M \to \infty$), where $k^\infty$ is the optimal coupling. The third equality in Eq. (40) is based on the duality of the EOT problem (Peyré et al., 2019) and $f^c$ is the soft c-transform defined as

$$f^c(\boldsymbol{x}') = -\log\Big(\int_{\mathbb{R}^D} \exp(\log k^0(\boldsymbol{x}, \boldsymbol{x}') + f(\boldsymbol{x}))\mathrm{d}\mu(\boldsymbol{x})\Big). \tag{41}$$

In such a situation, we can follow the proof in (Sander et al., 2022): It has been known that when $f$ is optimized, we have $f = f^c$, and $k^\infty(\boldsymbol{x}, \boldsymbol{x}') = \exp(\log k^0(\boldsymbol{x}, \boldsymbol{x}') + f(\boldsymbol{x}) + f(\boldsymbol{x}'))$. As a result, we have

$$\begin{aligned}
\nabla_W E^\infty(\mu)(\boldsymbol{x}) &= \nabla_{\boldsymbol{x}} f(\boldsymbol{x}) \\
&= \int_{\mathbb{R}^D} \exp(f(\boldsymbol{x}) + f(\boldsymbol{x}') + \log k^0(\boldsymbol{x}, \boldsymbol{x}'))\nabla_{\boldsymbol{x}} \log k^0(\boldsymbol{x}, \boldsymbol{x}')\mathrm{d}\mu(\boldsymbol{x}') \\
&= \int_{\mathbb{R}^D} k^\infty(\boldsymbol{x}, \boldsymbol{x}')\nabla_{\boldsymbol{x}} \log k^0(\boldsymbol{x}, \boldsymbol{x}')\mathrm{d}\mu(\boldsymbol{x}') \\
&= \int_{\mathbb{R}^D} k^\infty(\boldsymbol{x}, \boldsymbol{x})\boldsymbol{A}\boldsymbol{x}'\mathrm{d}\mu(\boldsymbol{x}') \\
&= -T_\mu^\infty(\boldsymbol{x}).
\end{aligned} \tag{42}$$

$\square$

## B.2. Proof of Proposition 4.2

*Proof.* Given non-Dirac measures $\{\mu_i\}_{i=1}^N$, we can fix each component measure except $\mu_i$ and minimize the energy function in Eq. (19), i.e., $\min_{\mu_i} E(\mu_i; \{\mu_j\}_{j\neq i})$. According to the work in (Santambrogio, 2017), the first variation of $E(\mu_i; \{\mu_j\}_{j\neq i})$ at $\mu_i$ is

$$\begin{aligned}
\frac{\delta E}{\delta \mu_i}(\mu_i)(\boldsymbol{x}) =& \frac{1}{2} \int_{\mathcal{X}} \exp(\boldsymbol{x}^\top \boldsymbol{A}\boldsymbol{x}' + r_{ii}) + \exp(\boldsymbol{x}'^\top \boldsymbol{A}\boldsymbol{x} + r_{ii})\mathrm{d}\mu_i(\boldsymbol{x}') + \\
& \frac{1}{2} \sum_{j\neq i} \int_{\mathcal{X}} \exp(\boldsymbol{x}^\top \boldsymbol{A}\boldsymbol{x}' + r_{ij}) + \exp(\boldsymbol{x}'^\top \boldsymbol{A}\boldsymbol{x} + r_{ji})\mathrm{d}\mu_j(\boldsymbol{x}') \\
=& \sum_{j=1}^N \int_{\mathcal{X}} \exp(\boldsymbol{x}^\top \boldsymbol{A}\boldsymbol{x}' + r_{ij})\mathrm{d}\mu_j(\boldsymbol{x}'),
\end{aligned} \tag{43}$$

where the second equality in Eq. (43) is because both $\boldsymbol{A}$ and $\boldsymbol{R} = [r_{ij}]$ are symmetric matrices. By differentiation under the integral, we have

$$\begin{aligned}
\nabla_W E(\mu_i; \{\mu_j\}_{j\neq i})(\boldsymbol{x}) &= \nabla\Big(\frac{\delta E}{\delta \mu_i}(\mu_i)\Big)(\boldsymbol{x}) \\
&= \sum_{j=1}^N \int_{\mathcal{X}} \nabla_{\boldsymbol{x}} \exp(\boldsymbol{x}^\top \boldsymbol{A}\boldsymbol{x}' + r_{ij})\mathrm{d}\mu(\boldsymbol{x}') \\
&= \sum_{j=1}^N \int_{\mathcal{X}} \exp(\boldsymbol{x}^\top \boldsymbol{A}\boldsymbol{x}' + r_{ij})\boldsymbol{A}\boldsymbol{x}'\mathrm{d}\mu(\boldsymbol{x}') \\
&= -T_{\mu_i}^0(\boldsymbol{x}).
\end{aligned} \tag{44}$$

$\square$

Table 5. The scale overview of Molecule3D and QM9 datasets.

| Dataset | Splitting | Training | Validation | Test |
|---|---|---|---|---|
| Molecule3D | random-splitting | 2,339,745 | 779,918 | 779,915 |
| | scaffold-splitting | 2,339,724 | 779,929 | 779,928 |
| QM9 | default | 108,831 | 9,900 | 10,697 |

### B.3. The Details of Eq. (17)

Note that, $\boldsymbol{X} \in \mathbb{R}^{N \times D}$, $\boldsymbol{W} \in \mathbb{R}^{D \times D}$, and $\boldsymbol{A} = \boldsymbol{W}\boldsymbol{W}^\top \in \mathbb{R}^{D \times D}$. Denote $\boldsymbol{Y} = \boldsymbol{X}\boldsymbol{W} \odot \boldsymbol{X}\boldsymbol{W}$, where $\odot$ represents the Hadamard product of matrix. The details of Eq. (17) is shown below:

$$
\begin{aligned}
\boldsymbol{P}^* &= \arg \max_{\boldsymbol{P} \in \Pi_1} \langle \boldsymbol{D}_{\boldsymbol{X}}, \boldsymbol{P} \rangle - \langle \boldsymbol{P}, \log \boldsymbol{P} \rangle \\
&= \arg \min_{\boldsymbol{P} \in \Pi_1} \langle -\boldsymbol{D}_{\boldsymbol{X}}, \boldsymbol{P} \rangle + \langle \boldsymbol{P}, \log \boldsymbol{P} \rangle \\
&= \arg \min_{\boldsymbol{P} \in \Pi_1} \langle \boldsymbol{R} - \frac{1}{2}(\boldsymbol{Y}\boldsymbol{1}_{D \times N} + \boldsymbol{1}_{N \times D}\boldsymbol{Y}^\top - 2\boldsymbol{X}\boldsymbol{W}\boldsymbol{W}^\top\boldsymbol{X}^\top), \boldsymbol{P} \rangle + \langle \boldsymbol{P}, \log \boldsymbol{P} \rangle \\
&= \arg \min_{\boldsymbol{P} \in \Pi_1} \langle \boldsymbol{X}\boldsymbol{A}\boldsymbol{X}^\top + \boldsymbol{R}, \boldsymbol{P} \rangle + \langle \boldsymbol{P}, \log \boldsymbol{P} \rangle \underbrace{- \frac{1}{2}\langle \boldsymbol{Y}\boldsymbol{1}_{D \times N}, \boldsymbol{P} \rangle - \frac{1}{2}\langle \boldsymbol{1}_{N \times D}\boldsymbol{Y}^\top, \boldsymbol{P} \rangle}_{\text{A constant due to the doubly stochastic constraint}} \\
&= \arg \min_{\boldsymbol{P} \in \Pi_1} \langle \boldsymbol{X}\boldsymbol{A}\boldsymbol{X}^\top + \boldsymbol{R}, \boldsymbol{P} \rangle + \langle \boldsymbol{P}, \log \boldsymbol{P} \rangle,
\end{aligned}
\tag{45}
$$

where $\langle \boldsymbol{Y}\boldsymbol{1}_{D \times N}, \boldsymbol{P} \rangle = \text{tr}(\boldsymbol{Y}\boldsymbol{1}_{D \times N}\boldsymbol{P}^\top) = \text{tr}(\boldsymbol{Y}\boldsymbol{1}_{D \times N})$ because $\boldsymbol{P} \in \Pi_1$.

## C. More Experimental Details and Results

### C.1. Datasets

As mentioned in the main body, we also employ the widely used Molecule3D and QM9 datasets in our work to remain consistent with previous works, thus evaluating the performance of our method fairly. The introductions of the above two datasets are provided below:

- **Molecule3D.** The first benchmark introduced by (Xu et al., 2021d) for the molecular ground-state conformation prediction task. It comprises approximately 4 million molecules, each with the corresponding ground-state conformation determined through density functional theory (DFT) calculations. Additionally, it provides two dataset-splitting results (i.e., random-splitting and scaffold-splitting) to ensure a comprehensive evaluation.

- **QM9.** A small-scale quantum chemistry dataset proposed in (Ramakrishnan et al., 2014), and the work in (Xu et al., 2023) further applies it to the molecular ground-state conformation prediction task. It comprises approximately 130,000 molecules with up to 9 heavy atoms, each with the corresponding ground-state conformation determined through density functional theory (DFT) calculations. Besides, it adopts the dataset-splitting strategy in (Liao & Smidt, 2023).

Table 5 provides a scale overview of Molecule3D and QM9 datasets, outlining their division into training, validation, and test sets. For the Molecule3D dataset, both random-splitting and scaffold-splitting strategies are applied, resulting in about 2.3 million molecules for training and approximately 0.78 million for both validation and test sets. Meanwhile, the QM9 dataset utilizes the default splitting strategy described in (Liao & Smidt, 2023), consisting of 108,831 molecules in the training set, 9,900 in the validation set, and 10,697 in the test set.

In addition, Figure 7 further visualizes the distribution of atom count (including the hydrogen atom) across these datasets. It demonstrates that molecules in the Molecule3D dataset generally contain more atoms than those in the QM9 dataset, thereby increasing the difficulty of predicting their corresponding ground-state molecular conformations. This observation aligns with the actual performance presented in Table 1, thus explaining the decreased performance of all methods on the Molecule3D dataset relative to the QM9 dataset.

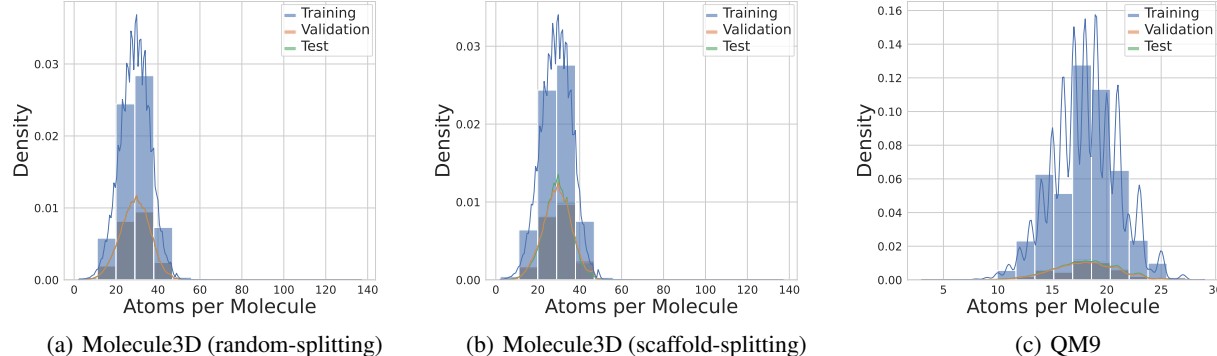

(a) Molecule3D (random-splitting)      (b) Molecule3D (scaffold-splitting)      (c) QM9

*Figure 7.* The atom count distribution of Molecule3D and QM9 datasets.

### C.2. Evaluation

As mentioned in the main body, to guarantee a comprehensive and fair evaluation, we choose the same metrics introduced by (Xu et al., 2023), including mean-absolute-error of distances (D-MAE), root-mean-squared-error of distances (D-RMSE), and root-mean-square-deviation of coordinates (C-RMSD), to evaluate the performance of various models. Here, the specific definitions of these evaluation metrics can be expressed as follows:

$$\text{D-MAE}(\widehat{\mathcal{D}}, \mathcal{D}^*) = \frac{1}{N^2} \sum_{i=1}^{N} \sum_{j=1}^{N} |\hat{d}_{ij} - d_{ij}^*|, \tag{46}$$

$$\text{D-RMSE}(\widehat{\mathcal{D}}, \mathcal{D}^*) = \sqrt{\frac{1}{N^2} \sum_{i=1}^{N} \sum_{j=1}^{N} (\hat{d}_{ij} - d_{ij}^*)^2}, \tag{47}$$

$$\text{C-RMSD}(\widehat{\mathcal{C}}, \mathcal{C}^*) = \sqrt{\frac{1}{N} \sum_{i=1}^{N} \|\hat{\boldsymbol{c}}_i - \boldsymbol{c}_i^*\|_2^2}. \tag{48}$$

where $\widehat{\mathcal{C}} = [\hat{\boldsymbol{c}}_i] \in \mathbb{R}^{N \times 3}$ is predicted ground-state conformation (zero-mean), $\mathcal{C}^* = [\boldsymbol{c}_i^*] \in \mathbb{R}^{N \times 3}$ is the ground-truth aligned through Eq. (20), $N$ is the number of atoms in the molecule, and $\hat{d}_{ij} = \|\hat{\boldsymbol{c}}_i - \hat{\boldsymbol{c}}_j\|_2$, $d_{ij}^* = \|\boldsymbol{c}_i^* - \boldsymbol{c}_j^*\|_2$.

### C.3. Baselines

As mentioned in the main body, we compare our method with 2D methods that predict ground-state conformations merely based on 2D molecular graphs and 3D methods that predict ground-state conformations from the conformation optimization perspective to validate the effectiveness of our method. The introductions of typical baselines are provided below:

- **GTMGC**: The typical 2D method proposed in (Xu et al., 2023), which introduces a novel network based on Graph-Transformer (GT) to predict the molecular ground-state conformation from the corresponding 2D molecular graph in an end-to-end manner. In addition, it further designs various 2D baselines by replacing the backbone module with other models like **GINE** (Hu et al., 2020), **GATv2** (Brody et al., 2021), and **GPS** (Rampášek et al., 2022).

- **ConfOpt**: The typical 3D method proposed in (Guan et al., 2022), which introduces variants of SE(3)-equivariant neural networks to predict the gradient field of the conformational energy landscape, and then optimize the input conformation by gradient descent. It can be divided into two variants: ConfOpt-TwoAtom, which considers only the relationships between atom pairs, and ConfOpt-ThreeAtom, which additionally accounts for the relationships among atom triplets. In addition, it further designs various 3D baselines by replacing the backbone module with other models like **SE(3)-Transformer** (Fuchs et al., 2020) and **EGNN** (Satorras et al., 2021).

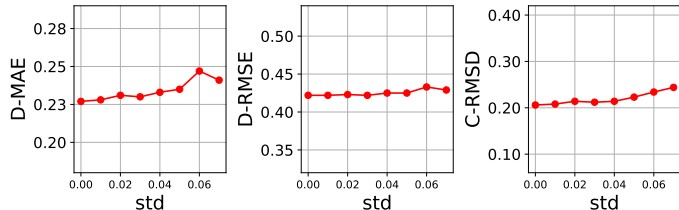

*Figure 8.* The robustness analysis of WGFormer to data noise on QM9.

*Table 6.* The specific hyperparameters on Molecule3D and QM9 datasets.

| Hyperparameter | Molecule3D | | QM9 |
| --- | --- | --- | --- |
| | random-splitting | scaffold-splitting | |
| Batchsize | 24 | 32 | 32 |
| Learning Rate | 5e-6 | 1e-5 | 2e-5 |
| Auxiliary Loss Weight ($\lambda$) | 0.125 | 0.125 | 0.125 |
| Epoch | 500 | 500 | 500 |
| Adam Eps | 1e-6 | 1e-6 | 1e-6 |
| Adam Betas | (0.9, 0.99) | (0.9, 0.99) | (0.9, 0.99) |
| Num Encoder Layer ($L$) | 30 | 30 | 30 |
| Relational Representation Dimension ($H$) | 64 | 64 | 64 |
| Atom-level Representation Dimension ($D$) | 512 | 512 | 512 |
| Attention Head Dimension ($D_a$) | 8 | 8 | 8 |

*Table 7.* The performance of WGFormer with repeated layers on QM9.

| Model | D-MAE $\downarrow$ | D-RMSE $\downarrow$ | C-RMSD $\downarrow$ |
| --- | --- | --- | --- |
| GTMGC | 0.264 | 0.470 | 0.367 |
| ConfOpt | 0.244 | 0.438 | 0.246 |
| WGFormer (30 different layers) | 0.227 | 0.422 | 0.206 |
| WGFormer (6 repeated layers) | 0.231 | 0.422 | 0.219 |

## C.4. Hyperparameters

As mentioned in the main body, our model comprises 30 encoder layers (i.e., the number of WGFormer modules), each equipped with 64 attention heads. Meanwhile, the atom-level and relational representation dimensions are set to 512 and 64, respectively. In addition, we train the whole model on a single NVIDIA RTX 3090 GPU and select the optimal checkpoint based on performance evaluated on the corresponding validation set. Here, the specific hyperparameters used across all datasets are summarized in Table 6. Please note that the batch size on the Molecule3D dataset (i.e., 24 for random-splitting and 32 for scaffold-splitting) is purely decided by GPU memory limitation in our work.

## C.5. More Experimental Results

**Robustness analysis.** Given the trained model on QM9, we add Gaussian noise to the input conformations during the inference phase to evaluate our model's robustness. As shown in Figure 8, our model achieves stable performance across various noise levels, demonstrating its robustness to data noise. The robustness of our model ensures reliable molecular ground-state conformation prediction even in the presence of perturbations.

**Performance with repeated layers.** In addition to training our model with 30 different WGFormer layers (i.e., the one in Table 1), we also train another model on QM9, which contains only 6 different WGFormer layers and repeats each layer 5 times. In this case, the repetition of each layer can be interpreted as optimizing the same energy functional with 5 Euler steps. As shown in Table 7, this model using repeated layers leads to comparable performance, thus further validating and supporting our theoretical analysis in Section 4.

*Table 8.* The median absolute errors of various property metrics on QM9.

| Model | $E$ (kcal/mol) | $\mu$ (debye) | $\Delta\epsilon$ (kcal/mol) |
|---|---|---|---|
| GTMGC | 0.008 | 0.014 | 0.068 |
| ConfOpt | 0.006 | 0.012 | 0.069 |
| WGFormer (ours) | **0.004** | **0.009** | **0.053** |

*Table 9.* The performance and efficiency comparisons with generative models on QM9.

| Model | D-MAE↓ | D-RMSE↓ | C-RMSD↓ | Inference Time (s/mol) |
|---|---|---|---|---|
| GeoDiff | 0.278 | 0.563 | 0.473 | 18.058 |
| TorsionDiff | 0.378 | 0.685 | 0.437 | 14.359 |
| WGFormer (ours) | **0.227** | **0.422** | **0.206** | **0.150** |

*Table 10.* The performance and efficiency comparisons on DRUGS.

| Model | D-MAE↓ | D-RMSE↓ | C-RMSD↓ | Inference Time (s/mol) |
|---|---|---|---|---|
| GTMGC | 0.914 | 1.420 | 1.657 | 1.878 |
| ConfOpt | 0.825 | **1.272** | 1.650 | 48.135 |
| WGFormer (ours) | **0.816** | 1.281 | **1.602** | **0.778** |

**Chemical property metrics.** Following the pipeline established in (Jing et al., 2022), we further compare the chemical properties of the predicted and reference ground-state conformations to evaluate the performance of our WGFormer comprehensively. In particular, given each trained model on QM9, we employ the xTB tool (Bannwarth et al., 2019) to compute three popular chemical properties (energy $E$, dipole moment $\mu$, HOMO-LUMO gap $\Delta\epsilon$) and calculate the median absolute errors on the test set. As shown in Table 8, our WGFormer significantly outperforms GTMGC (the best 2D baseline) and ConfOpt (the best 3D baseline) across all property metrics, further validating its superiority.

**Comparisons with generative models.** While existing generative models are originally designed to generate multiple conformations, we preserve the one with the lowest energy as the predicted ground-state conformation for comparisons. Here, we train two typical generative models (i.e., GeoDiff (Xu et al., 2021c) and TorsionDiff (Jing et al., 2022)) on QM9 and compare them with our WGFormer on the test set. As shown in Table 9, our WGFormer significantly outperforms the above two generative models across all evaluation metrics, achieving lower prediction errors and higher inference speed. This remarkable result highlights the practicality and superiority of our WGFormer for the molecular ground-state conformation prediction task, where both accuracy and efficiency are critical.

**Comparisons on the DRUGS dataset.** Since each molecule in the original DRUGS dataset has multiple conformations, and we don't know which one is the ground-state conformation (or even whether the ground-state conformation is among them), this dataset is not well suited for the molecular ground-state conformation prediction task. To make this dataset applicable and further train various models, we select the conformation with the highest Boltzmann weights as the ground-state conformation and adopt the identical dataset-splitting strategy described in (Xu et al., 2021c). As shown in Table 10, compared with GTMGC (the best 2D baseline) and ConfOpt (the best 3D baseline), our WGFormer significantly outperforms them, achieving state-of-the-art performance with the highest inference speed. This demonstrates the generalizability of our WGFormer across various datasets, thus enabling it to meet the requirements of real-world applications effectively.

### C.6. More Cases about Energy Change

In Figure 9, we have provided more cases about the change of latent and potential energy as the number of layers increases, where latent energy is obtained by solving Eq. (17) and potential energy is obtained by the xTB tool (Bannwarth et al., 2019). As shown in this figure, both latent and potential energy decrease gradually as the number of layers increases, and their change exhibit remarkably high correlation. These results validate that minimizing this latent energy can drive potential energy minimization, thereby being highly correlated with conformation optimization.

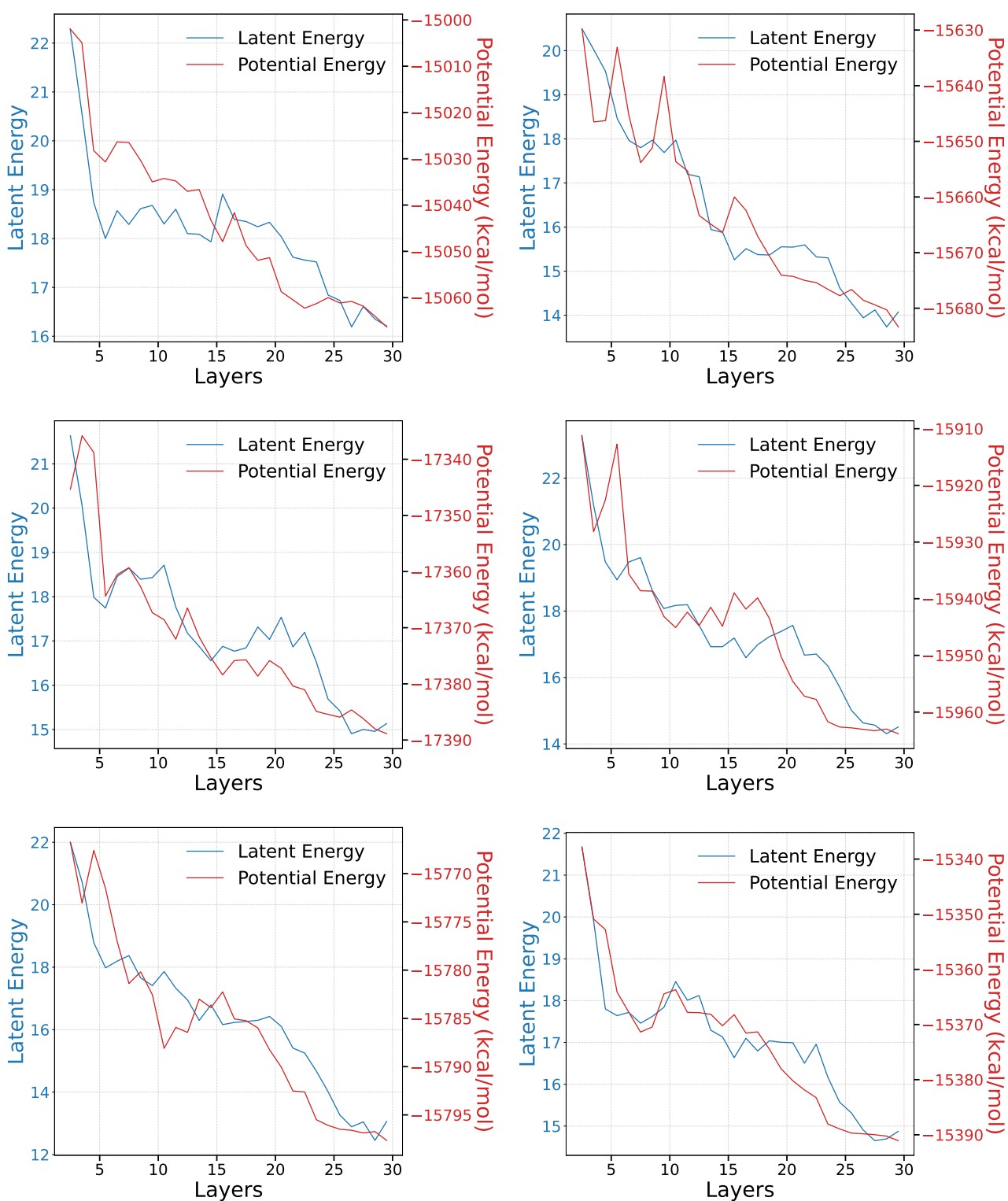

*Figure 9.* More cases about the change of latent and potential energy as the number of layers increases.

