# OpenReview forum: "WGFormer: An SE(3)-Transformer Driven by Wasserstein Gradient Flows for Molecular Ground-State Conformation Prediction"
_ICML.cc/2025/Conference — ICML 2025 poster_

### Official Review · Reviewer_vjqn · 2025-03-06

**Overall Recommendation:** 2

**Summary:**

This paper proposes a transformer-based neural network architecture to predict the ground state conformation of molecules from some low quality conformation (equivalent to optimization). The encoder part of the model takes as input per-atom embeddings (depending only on atom types) and atom-pair embeddings (distances after Gaussian kernel expansion with some learned weight), and applies the attention mechanism for multiple layers to generate the final representation for each atom. Finally, the decoder predicts deviation in pair-wise distances to update the coordinates of the input conformation.

**Claims And Evidence:**

- The first claim of this work is efficiency. "*Classic energy-based simulation is time-consuming when solving this problem while existing learning-based methods have advantages in computational efficiency but sacrifice accuracy and interpretability. In this work, we propose a novel and effective method to bridge the energy-based simulation and the learning-based strategy.*" This paper identifies WGFormer as a method that is faster than energy-based methods and more accurate than other learning-based methods. However, only accuracy is demonstrated and no discussion about efficiency and especially comparison to quantum-based methods has been provided.
- Another claim of this work is interpretability. This paper describes the forward pass of the encoder as a process of minimization of some energy function, and suddenly it comes to a conclusion that the latent energy function of this process "*is an interpretable energy function for conformation optimization*". This makes no sense as this paper didn't reveal at all how exactly the gradient flow process relates to the physical process of conformation optimization. It seems to me that the conclusion of interpretability is drawn only because both the W.G.flow and the conformation optimization are some kind of energy minimization processes.
- Further, in the Related Work section, the paper states that "*the empirical architectures of the models prevent them from minimizing a physically meaningful energy function, resulting in limited model interpretability and sub-optimal performance*" which is not true. For example, DSMBind [1], an application study of a conformation optimization model, shows that the model's output correlates with physical quantities (binding affinity), which gives interpretability.

[1] Jin, Wengong, et al. "Unsupervised protein-ligand binding energy prediction via neural euler's rotation equation." Advances in Neural Information Processing Systems 36 (2023): 33514-33528.

**Essential References Not Discussed:**

N/A

**Experimental Designs Or Analyses:**

There is no major issue with experimental designs.

**Methods And Evaluation Criteria:**

There is no major issue with the design of the network. The motivation of getting inspiration from Sinkformer and the three improvement techniques is unclear, though supported by ablation studies.

**Other Comments Or Suggestions:**

N/A

**Other Strengths And Weaknesses:**

N/A

**Questions For Authors:**

N/A

**Relation To Broader Scientific Literature:**

The work focuses on small molecule conformation optimization. There have been a lot of papers in this area. This paper is not very special and does not make unique contribution.

**Theoretical Claims:**

The main theoretical claim is the relationship between the WG flow and the conformational energy minimization process, which is not grounded.

---

> ### Author Rebuttal · Authors · 2025-03-31
>
> For your comments, below we answer them one by one.
>
> **W1: No discussion about efficiency**
>
> **A1:** Firstly, the inefficiency of energy-based/quantum-based methods (e.g., molecular dynamics simulation and density functional theory calculation) has already been a well-established consensus (also has been discussed in lines 12-20 of our paper) in this field [1][2][3], which is why numerous learning-based methods are proposed to approximate their accuracy.
>
> Secondly, as shown in Figure 4 of our paper, we have demonstrated that our WGFormer is significantly faster than existing learning-based methods, reducing the runtime per molecule by over 50%.
>
> **W2&W5: Lack interpretability and the relationship with the conformational energy minimization process**
>
> **A2&W5:** As we have mentioned in lines 30-35, our WGFormer is a Wasserstein gradient flow-driven SE(3)-Transformer architecture. It optimizes molecular conformations by **minimizing an energy function (i.e., Eq.17) defined on the latent mixture models of atoms.** The corresponding proofs, including the relationship between our energy function and conformation optimization (lines 220-242), have been provided in Section 4. To further verify our claim, we have conducted analytic experiments in our response to Reviewer dW7c (https://openreview.net/forum?id=2wUQttiab3&noteId=bSAlOuwM9f).
>
> These experiments demonstrate that:
>
> 1) Our WGFormer is indeed minimizing the energy function defined on the latent mixture models of atoms;
>
> 2) Minimizing this energy function indeed helps optimize the physically-meaningful energy of molecular conformation, and these two kinds of energy are highly correlated;
>
> 3) Minimizing this energy function is indeed closely relevant to improving the final metrics.
>
> **W3: Clarification of relevant statements**
>
> **A3:** The interpretability we emphasize here refers to the fact that the feedforward computation of existing model architectures (e.g., those baselines) cannot effectively correspond to the process of molecular conformation optimization. In contrast, WGFormer optimizes molecular conformations by minimizing an energy function defined on the latent mixture models of atoms, whose feedforward computation is the Euler step solving a continuity equation.
>
> The DSMBind you mentioned works to predict protein-ligand binding energy, which is unrelated to our work in either problem setting or technical route.
>
> **W4: Unclear motivation**
>
> **A4:** The motivation of our model design is clear --- as shown in lines 66-85, we aim to build a new SE(3)-equivalent model with interpretable architecture for obtaining molecular ground-state conformations from their low-quality conformations. To achieve this aim, we formulate the task as a conformation energy optimization problem, so that the feedforward computation of the model needs to be the Wasserstein gradient flow of the contunity equation in Eq.13. All our architectural improvements, including modifying QKV mechanism and applying Sinkhorn-based attention (as shown in lines 206-219), serve for this aim.
>
> **W6: A lot of same work and lack contribution**
>
> **A6:** We respectfully disagree with this viewpoint. In fact, all other reviewers have widely recognized the contribution of our work, including interesting and important relations with energy-based methods (Reviewer dW7c), the first attempt to predict molecular ground-state conformation through the lens of Wasserstein gradient flow (Reviewer NScR), and the fact that performance improvements are sound and lean in a promising direction in terms of architectural exploration (Reviewer FN8S).
>
> **In the aspect of model architecture,** we propose a first Wasserstein gradient flow-driven SE(3)-Transformer for molecular modeling. It is the first SE(3)-equivariant model that corresponds to a Wasserstein gradient flow in the latent space of atoms, which not only enhances the model interpretability but also improves computational efficiency.
>
> **In the aspect of theory,** we build the connection between the model architecture and the Wasserstein gradient flow and provide the conditions for the connection. In addition, as shown in Eq.17, we analyze the latent energy function in depth and explain its rationality from the perspective of entropic OT.
>
> **In the aspect of application,** currently, few attempts are made to predict molecular ground-state conformations. Our work provides a strong and interpretable solution to this challenging and important problem.
>
> In summary, from any perspective, we believe our work deserves a higher score. We respectfully ask you for re-evaluating our work in light of the responses above and the comments of the other reviewers.
>
> [1] Learning gradient fields for molecular conformation generation, ICML 2021.
>
> [2] Learning neural generative dynamics for molecular conformation generation. arXiv preprint 2021.
>
> [3] Energy-annotated molecular conformations for property prediction and molecular generation. Scientific Data, 2022.

---

### Official Review · Reviewer_FN8S · 2025-03-12

**Overall Recommendation:** 3

**Summary:**

This paper proposes a transformer architecture for optimizing molecular geometries. The network first processes node features and edge features (encoding the input geometry) via residual update blocks with a bespoke attention mechanism called WGFormer. An interpretation of the WGFormer is provided as a Wasserstein gradient flow in latent space. These features are then used to update the input 3D geometry using relative position vectors. The model is shown to outperform existing methods on this task in terms of MAE and RMSD metrics.

**Claims And Evidence:**

The main claim of improved model performance is supported. The results on QM9 are to my knowledge SOTA, but it would be nice to see results on larger and more diverse datasets, e.g., DRUGS.

The claim that the WGFormer module, and its interpretation as Wasserstein gradient flow, is responsible for this improved performance is significantly less convincing.
*    The attention based module with explicitly updated dense pair representations bears some similarity to protein structure prediction networks; it is perhaps not so surprising that such an architecture would by itself already outperform existing methods for predicting small molecule structures. Indeed, the paper's ablation results show that even without the Sinkhorn module, the model is already SOTA.
*    Although the Sinkhorn module helps, the interpretation as Wasserstein gradient flow is unconvincing. First of all, the interpretation only holds in the limit of infinitesimal updates, when residual blocks are reduced to a neural ODE. Second of all, the energy functional depends on $k$ and is therefore changing for every layer.
*    The analysis in Figure 5 merely shows that adding layers to the network helps performance, which is independent of the Wasserstein gradient flow interpretation. It would be more convincing if the authors could show that the behavior of non-WGFormers are different.
*    I should stress that an interpretation of the architecture improvement is not necessary to be a strong paper, but if the authors choose to feature it prominently, then it should be strongly supported.

**Essential References Not Discussed:**

I am not aware of any essential references not discussed.

**Experimental Designs Or Analyses:**

Please see comments in "Claims and Evidence."

**Methods And Evaluation Criteria:**

Yes, the benchmarks are standard. However, larger molecules (DRUGS) have been used in previous works, but are missing here. Also, it would be helpful to show chemical property metrics that are of interest to those computing properties from minimized conformers.

**Other Comments Or Suggestions:**

Despite the concerns about the interpretation of WGFormer, I lean towards accept because the performance improvements are sound and lean in the right direction in terms of architectural exploration.

**Other Strengths And Weaknesses:**

No additional comments

**Questions For Authors:**

No additional questions

**Relation To Broader Scientific Literature:**

This paper contributes to the ML literature on learning small molecule geometry optimizers. Although a coherent body of work at ML venues, the broader impact in actual scientific software is not yet clear. In this respect, this submission probably does move the needle much as the improvements are on the same metrics and are incremental (i.e., it is not clear where the threshold for wider applicability is, so it is not clear if it has been reached).

The paper seems to draw heavy inspiration from Sinkformer, with which I am less familiar. From the point of view of other areas of AI4Science though, the proposed architecture, with its dense pair features used to bias attention, resembles architectures commonly used for proteins. It could be quite interesting to explore further architectural developments along these lines, even if they do not admit clean theoretical interpretations.

**Theoretical Claims:**

I did not carefully check the proofs for theoretical claims.

---

> ### Author Rebuttal · Authors · 2025-03-31
>
> Thanks for your positive feedback and constructive comments. Below, we resolve your concerns one by one.
>
> **W1: Test on larger datasets.**
>
> **A1:** As shown in Table 1 of our paper, in addition to QM9, our WGFormer also achieves SOTA performance on the Molecule3D dataset, which contains about four million molecules and is larger than DRUGS. In addition, each molecule in DRUGS has multiple conformations, and we don't know which one is the ground-state conformation (even don't know whether the ground-state conformation is among them). So, this dataset is not very suitable for our task.
>
> Nevertheless, we follow your suggestion, adding an experiment on DRUGS. We select the conformation with the highest Boltzmann weights as its ground-state conformation, then train and test different models. The results below demonstrate WGFormer's superiority in accuracy and efficiency on DRUGS.
>
> |Method|D-MAE|D-RMSE|C-RMSD|Inference Time (s/mol)|
> |-|-|-|-|-|
> |GTMGC|0.914|1.420|1.657|1.88|
> |ConfOpt|0.825|**1.272**|1.650|48.14|
> |WGFormer|**0.816**|1.281|**1.602**|**0.78**|
>
> **W2: The novelty and significance of proposed model architecture**
>
> **A2:** Although attention-based models have been widely used in molecule and protein generation, our WGFormer is still valuable because of the following reasons:
>
> Firstly, as shown in Figure 2 and lines 206-214, WGFormer introduces new QKV architecture, leading to a new cross-head interaction mechanism with fewer model parameters. Secondly, as shown in Appendix A, WGFormer applies the Sinkhorn-based attention module while maintaining the SE(3)-equivariance property, which enhances the interpretability of the model and makes it suitable for 3D molecular modeling. Combining these two improvements jointly, WGFormer can be interpreted as Wasserstein gradient flow for the latent mixture model of atoms.
>
> No matter whether the original SE(3)-Transformer works well or not, our architectural improvements have led to better performance with fewer trainable parameters and better interpretability, demonstrating the rationality of our design.
>
> **W3:  1) Can finite Sinkhorn iterations be interpreted as Wasserstein gradient flow? 2) The energy functional depends on $k$ and changes for every layer.**
>
> **A3:** **Firstly, using finite Sinkhorn iterations is not strong evidence that the model cannot be interpreted as Wasserstein gradient flow.** The gap between theoretical analysis and practical implementation is natural --- for convex optimization, we stop an algorithm in finite iterations, but it does not mean its analysis based on infinite series is meaningless. In practice, the Sinkhorn algorithm converges very fast to the optimum. The Sinkformer in [2] merely applies 3-5 Sinkhorn iterations. Our WGFormer follows the same setting.
>
> **Secondly, $k$ changes do not mean the energy functional changes for every layer.** In our paper, the energy functional is shown in Eq.(17), which is formulated as an entropic OT problem. Each WGFormer layer optimizes the same energy functional, leading to the Wasserstein gradient flow (lines 220-251, left column).
>
> **W4&W5:  Interpretations of architecture improvement and Figure 5**
>
> **A4&A5:** Our WGFormer can optimize molecular conformations by minimizing an energy function defined on the latent mixture models of atoms. To further verify our claim, we have conducted additional experiments in our response to Reviewer dW7c (https://openreview.net/forum?id=2wUQttiab3&noteId=bSAlOuwM9f). These experiments demonstrate that:
>
> 1) WGFormer indeed minimizes a latent energy function for atoms' probability measure;
>
> 2) Minimizing this latent energy helps optimize the physical energy of conformation, highly correlated with the final metrics.
>
> For Figure 5, as shown in lines 426-439, we first train a WGFormer with $L=30$ layers and test it using different layer numbers ($L=1,...,30$). **The performance is improved as the number of layers increases, indicating that each layer is an Euler step to minimize the energy function in Eq.17.** When applying a non-WGFormer architecture with $L$ layers, the performance may not be improved consistently when increasing the number of layers, as shown in this anonymous link (https://anonymous.4open.science/r/WGFormer-comparison/Non-WGFormer.pdf).
>
> **W6: Chemical property metrics**
>
> **A6:** Following [1], we compare predicted and real ground-state conformations on their energy (in kcal/mol), dipole moment (in debye), and HOMO-LUMO gap (in kcal/mol). The MAEs of WGFormer and typical baselines are shown below, further demonstrating WGFormer's superiority.
>
> |Method|Energy|Dipole Moment|HOMO-LUMO Gap|
> |-|-|-|-|
> |GTMGC|0.008| 0.014|0.068|
> |ConfOpt|0.006|0.012|0.069|
> |WGFormer|**0.004**|**0.009**|**0.053**|
>
> Hope the above responses help enhance your confidence to further support our work.
>
> [1] Torsional diffusion for molecular conformer generation, NeurIPS 2022.
>
> [2] Sinkformer: Transformers with doubly stochastic attention, AISTATS 2022.

---

> > ### Comment · Reviewer_FN8S · 2025-04-07
> >
> > "Secondly, changes do not mean the energy functional changes for every layer."
> > What is $A$ in Eq 17? If $A=WW^T$ as previously defined then this changes for every layer with weights $W$.
> >
> > "The performance is improved as the number of layers increases, indicating that each layer is an Euler step to minimize the energy function in Eq.17."
> > Could the authors please clarify exactly how this experiment is carried out? There are many possible interpretations, some of which are consistent with the authors' claims, and some less so.
> >
> > "Firstly, using finite Sinkhorn iterations is not strong evidence that the model cannot be interpreted as Wasserstein gradient flow. The gap between theoretical analysis and practical implementation is natural --- for convex optimization, we stop an algorithm in finite iterations, but it does not mean its analysis based on infinite series is meaningless"
> > I believe the authors here are conflating the difference between truncation and discretization. My criticism holds analogously to the interpretation of residual updates as neural ODEs --- the discretization introduces qualitatively different behavior. For example, neural ODEs are always invertible whereas residual networks are nearly never so.

---

> > > ### Author Response · Authors · 2025-04-08
> > >
> > > Thanks for your feedback. Below, we try to resolve your remaining concerns one by one.
> > >
> > > **Q1: The energy function changes with respect to the weight $W$ in each layer.**
> > >
> > > **A1.** Sorry for misunderstanding your question. In our original rebuttal, we mean that given $A$, the function $k$ and the corresponding energy function are defined accordingly, no matter what kind of algorithm is applied to optimize the energy.
> > >
> > > We indeed learn different $W$'s for different layers in our experiment. However, this setting follows Sinkformer, which tries to interpret Transformer (not SE(3)-Transformer) from the perspective of Wasserstein gradient flow. Moreover, even if the energy function changes for different layers, each WGFormer layer can still be interpreted as one-step updating of the energy associated with the current layer.
> > >
> > > Nevertheless, we add the following experiment to resolve your concern. **Besides training WGFormer with 30 different layers, we train another WGFormer, which contains only 6 different layers and repeats each layer 5 times. In such case, the repeating of each layer can be interpreted as optimizing the same energy functional with 5 Euler steps.** Due to the limited rebuttal time, we train the model on QM9 and compare it with the baselines and the 30-layer WGFormer. The results below show that the WGFormer using repeated layers leads to comparable performance.
> > >
> > > ||D-MAE|D-RMSE|C-RMSD|
> > > |-|-|-|-|
> > > |GTMGC|0.264|0.470|0.367|
> > > |ConfOpt|0.244|0.438|0.246|
> > > |WGFormer (30 Layers)|0.227|0.422|0.206|
> > > |WGFormer (6 Repeated Layers)|0.231|0.422|0.219|
> > >
> > > In the revised paper, we will add this result and try WGFormer with fewer repeated layers.
> > >
> > > **Q2: Further explain the experiment obtaining Figure 5**
> > >
> > > **A2:** As we have mentioned in lines 426-439 of the paper and the response to Reviewer dW7c (https://openreview.net/forum?id=2wUQttiab3&noteId=bSAlOuwM9f), we first train a WGFormer with 30 layers through the defined loss function (i.e., Eq. 21), then we fix the model and use it to infer ground-state conformations in the testing set. During the inference, we obtain the interatomic relational representation $\mathbf{R}^{(l)}$ of each layer ($l$=1,...,30), and pass each of them through the trained decoder to obtain the molecular conformation (i.e., Eq.3) corresponding to each layer. For these molecular conformations obtained through different layers, we can further use the evaluation metrics (i.e., D-MAE, D-RMSE and C-RMSD) to measure how are these conformations close to the ground-state conformation.
> > >
> > > As shown in Figure 5, the metrics are improved as the number of layers increases. Besides, when we conduct this experiment on a non-WGFormer architecture with 30 layers, the metrics are often not improved consistently when increasing the number of layers, as shown in https://anonymous.4open.science/r/WGFormer-comparison/Non-WGFormer.pdf. Moreover, as shown in the response to Reviewer dW7c (https://openreview.net/forum?id=2wUQttiab3&noteId=bSAlOuwM9f), the reduction of the physical energy is highly correlated with the reduction of the latent energy in Eq.17 --- the Pearson correlation between them is larger than 0.9.
> > >
> > > In summary, this result demonstrates that WGFormer can be interpreted as the Euler step minimizing the latent energy defined in Eq.17, and accordingly, leads to the ground-state conformation optimization.
> > >
> > > **Q3: The rationality of interpreting the discretized feedforward steps as Wasserstein gradient flow.**
> > >
> > > **A3:** Thanks for further clarifying your concern. As shown in Section 4.1 and the above responses, we interpret each WGFormer layer as an Euler step (i.e., Eq.14) for solving the continuity equation in Eq.13. The equation describes the evolution of the latent mixture model of atoms in the time interval [0, 1] and the number of WGFormer layers corresponds to the number of Euler steps.
> > >
> > > We agree that the discretization may have different behaviors compared to its continuous counterpart. However, it should be noted that:
> > >
> > > 1) In practice, Euler method, although is discrete, is one of the most commonly used method to solve differential equations.
> > >
> > > 2) In theory, the sections 4.3 and 4.4 in the reference [1] (we cited in the paper) show that the time-discretized probability measure evolution (in the JKO scheme shown in Eq.(4.10)) converges to the Wasserstein gradient flow when the time step limits to infinitesimal (the content from Eq.(4.17) to Eq.(4.18)). In other words, the rationality of the discretized approximation is guaranteed in theory.
> > >
> > > Therefore, from the perspectives of theory and practice, interpreting WGFormer as an implementation of Wasserstein gradient flow, at least, has some rationality.
> > >
> > > **We hope the above responses can resolve your remaining concerns and enhance your confidence to further support our work in the decision phase.**
> > >
> > > [1] Santambrogio, Filippo. {Euclidean, metric, and Wasserstein} gradient flows: an overview. Bulletin of Mathematical Sciences 2017.

---

### Official Review · Reviewer_NScR · 2025-03-13

**Overall Recommendation:** 3

**Summary:**

The paper introduces WGFormer, a novel model that combines the strengths of energy-based simulation and learning-based methods for predicting molecular ground-state conformations. WGFormer is built upon an SE(3)-Transformer framework and is driven by Wasserstein gradient flows. In an auto-encoding setup, the model encodes low-quality 3D conformations (e.g., generated via RDKit) and decodes them into ground-state conformations using a lightweight MLP. A key innovation is the customized attention mechanism based on the Sinkhorn-scaling algorithm with adjusted QKV matrices and the omission of the feed-forward network, which together reduce computational cost and promote cross-head feature fusion. The theoretical framework establishes that, under certain conditions, WGFormer operates as a discretized version of Wasserstein gradient flows that minimize a physically meaningful energy function defined on a latent mixture model of atoms. Extensive experiments on datasets such as Molecule3D and QM9 show that WGFormer not only outperforms current state-of-the-art baselines in both accuracy (e.g., lower C-RMSD values) and efficiency but also demonstrates robustness through comprehensive ablation studies.

**Claims And Evidence:**

Yes

**Essential References Not Discussed:**

[1] Abramson, Josh, et al. "Accurate structure prediction of biomolecular interactions with AlphaFold 3." Nature 630.8016 (2024): 493-500.

**Experimental Designs Or Analyses:**

Yes

**Methods And Evaluation Criteria:**

- The comparision between WGFormer and other generative model (e.g., GeoDiff) is needed.

- There are many works optimize the low-quality molecule comformation to generate higher-quality molecules, such as AlphaFold3. It utilize the RDKit generated / CCD structures as reference features to predict the ligand structures. What are the difference between WGFormer and these methods?

- It seems that Wasserstein gradient needs the computation of gradient. What is the consumption of WGFormer? Adding the comparison with other generative model is preferred.

- What is the advantage of Wasserstein gradient flow compared with flow-matching based molecule generative model?

**Other Comments Or Suggestions:**

NA

**Other Strengths And Weaknesses:**

NA

**Questions For Authors:**

NA

**Relation To Broader Scientific Literature:**

This method makes the first attempt to predict molecular ground-state conformation through the lens of Wasserstein gradient flow.

**Theoretical Claims:**

Yes

---

> ### Author Rebuttal · Authors · 2025-03-31
>
> Thanks for your comments. Below, we resolve your concerns one by one.
>
> **Q1: Comparisons with other generative models**
>
> **A1:** Existing generative models generate multiple conformations by sampling. To make such models (e.g., GeoDiff [2] and TorsionDiff [3]) applicable for generating molecular ground-state conformations, we follow the commonly-used protocol in [1], generating multiple conformations and only preserving the one with the lowest energy.
>
> Using the above strategy, we train and test GeoDiff and TorsionDiff on QM9. The table below shows that WGFormer outperforms these two models, achieving lower prediction errors and much higher inference speed.
>
> |Method|D-MAE|D-RMSE|C-RMSD|Inference Time (s/mol)|
> |-|-|-|-|-|
> |GeoDiff|0.278|0.563|0.473|18.06|
> |TorsionDiff|0.378|0.685|0.437|14.36|
> |WGFormer|**0.227**|**0.422**|**0.206**|**0.15**|
>
> **Q2: The differences with other methods**
>
> **A2:** The differences between our WGFormer and the methods you mentioned can be summarized as follows:
>
> **1) Interpretable architecture:**
>
> As shown in lines 36-49, existing conformation optimization methods (e.g., ConfOpt-Two/Three Atoms) merely apply neural networks to approximate the gradients of atoms' motions when optimizing a molecule's conformation. The feedforward computations of their models do not correspond to minimizing a meaningful energy function. In contrast, our WGFormer is a Wasserstein gradient flow-driven SE(3)-Transformer. Its feedforward computation exactly corresponds to the Wasserstein gradient flow of the latent mixture model of atoms (i.e., the evolution of atoms' probability measure in the latent space). Each WGFormer layer is an Euler step minimizing an energy function of the molecular conformation, thereby significantly improving performance and interpretability.
>
> **2) Focus on ground-state conformation:**
>
> High-quality conformation $\neq$ Ground-state conformation. The AlphaFold series predicts protein structures but cannot ensure the structures are in the ground state. Although AlphaFold3 can predict biological complexes, it does not ensure ground-state conformations, either. WGFormer predicts the ground-state conformation, which determines basic molecular properties (shown in lines 43-48). Therefore, the AlphaFold series is not correlated with our work.
>
> **Q3: Does Wasserstein gradient need to compute gradient? The comparison on computational consumption is required.**
>
> **A3:**  Firstly, we would like to explain the key concepts clearly. As shown in lines 238-258, **Wasserstein gradient flow captures a probability measure's evolution in the Wasserstein space (e.g., the change of the latent mixture model of atoms over time) rather than the gradient of each atom.** The detailed differences between Wasserstein gradient flow and gradient flow can be found in [4], which has been cited in the paper.
>
> Secondly, **Wasserstein gradient flow is achieved by the model architecture itself rather than additional computation.** As shown in Section 4.1 and **A2**, the feedforward computation of WGFormer corresponds to the Euler step solving a continuity equation (i.e., Eq.10) and minimizes an energy function (i.e., Proposition 4.1).
>
> Thirdly, WGFormer improves SE(3)-Transformer's architecture without increasing the complexity. We can train it on a single 3090 GPU, and we have demonstrated its superiority on inference speed in Figure 4 of our paper and the Table in **A1**.
>
> **Q4: Comparison with flow-matching based methods?**
>
> **A4:** Our work, i.e., developing a Wasserstein gradient flow-based model, is different from the flow-matching learning strategy:
>
> **Theory:** Wasserstein gradient flow optimizes an energy function of probability measures in the Wasserstein space, while flow-matching aims to model the velocity field of particles in the sample space, which does not model the energy of particles or their distribution evolution explicitly.
>
> **Tech route:** Our work focuses on improving model architecture and making it interpretable as Wasserstein gradient flow. We do not change the model's learning paradigm. Flow-matching is a model-agnostic learning strategy for fitting the velocity field of data.
>
> **Implementation:** WGFormer is learned to predict the ground-state conformation, and its architecture ensures the feedforward computation leads to lower energy. Flow-matching methods focus on generating "valid" rather than energy-minimized conformations. In general, they cannot ensure the energy is minimized in the inference phase.
>
> Hope the above responses can resolve your concerns and help re-evaluate our work.
>
>
> [1] REBIND: Enhancing Ground-state Molecular Conformation Prediction via Force-Based Graph Rewiring, ICLR 2025.
>
> [2] Geodiff: A geometric diffusion model for molecular conformation generation, ICLR 2022.
>
> [3] Torsional diffusion for molecular conformer generation, NeurIPS 2022.
>
> [4] {Euclidean, metric, and Wasserstein} gradient flows: an overview. Bulletin of Mathematical Sciences, 2017.

---

> > ### Comment · Reviewer_NScR · 2025-04-06
> >
> > Thanks for the reply. I've read the authors' rebuttal, which have resolved my problems. I've raised my score.

---

> > > ### Author Response · Authors · 2025-04-06
> > >
> > > Dear Reviewer,
> > >
> > > Thanks for your valuable feedback!
> > > We are glad to hear that we have resolved your problems, and your generous decision to raise your score means a great deal to us.
> > > Your support will also help us continue our efforts in this field, and we would greatly appreciate it if you could continue to support our work in the following discussions.
> > >
> > > Thank you once more for your valuable contributions to enhancing our work.
> > >
> > > Best wishes,
> > >
> > > Authors

---

### Official Review · Reviewer_dW7c · 2025-03-16

**Overall Recommendation:** 4

**Summary:**

this work proposes "Wasserstein gradient flow-driven" transformer, to gradually refine a initial 3D conformation to its ground state.
this refinement process is associated with minimizing an energy function, thus enhancing its interpretability, and probably explains its performance improvement.
the method is validated with extensive experiments with several ablation studies.

**Claims And Evidence:**

yes for most claims.

still wondering about one core claim, i.e., "minimization of the energy function". several comments:
* could the author please visualize this process, e.g., plot the energy function value vs the layers, from input to output for illustration
* any way to verify this "energy function defined on the latent mixture models of atoms" is closely relevant to the final metrics?

**Essential References Not Discussed:**

N/A

**Experimental Designs Or Analyses:**

looks good to me

**Methods And Evaluation Criteria:**

looks good to me

**Other Comments Or Suggestions:**

see above

**Other Strengths And Weaknesses:**

see above

**Questions For Authors:**

see above

**Relation To Broader Scientific Literature:**

the relation with energy-based methods seem to be interesting and important to me.

**Theoretical Claims:**

I don't have enough expertise

---

> ### Author Rebuttal · Authors · 2025-03-31
>
> Thanks for your appreciation of our work. Below, we resolve your concerns one by one.
>
> **Q1: Visualize the latent energy function values achieved through different numbers of layers.**
>
> **A1:** As we have mentioned in lines 30-35 of our paper, our WGFormer is a Wasserstein gradient flow-driven SE(3)-Transformer architecture. In particular, it can optimize molecular conformations by minimizing an energy function defined on the latent mixture models of atoms (i.e., Eq.17 in the paper, denoted as `latent energy` for short), and the corresponding proofs have been provided in Section 4 of our paper.
>
> To further verify our claim, **given a trained WGFormer with 30 layers**, we have conducted a series of analytic experiments in the inference phase.
>
> Firstly, following your insightful suggestion, we randomly sample some RDKit-based molecular conformations and pass them through the trained WGFormers. Given the output (i.e., $\mathbf{X}$ and $\mathbf{R}$) of each layer, we can calculate the latent energy per layer by solving Eq.17 (using the Sinkhorn-scaling algorithm). Then, we plot the curve of the latent energy varying with the number of layers. As demonstrated by the figure in this anonymous link (https://anonymous.4open.science/r/WGFormer-energy/Latent_Energy.pdf), the latent energy decreases gradually as the number of layers increases, which strongly validates that our WGFormer is indeed minimizing the energy function defined on the latent mixture models of atoms.
>
> **Q2: Verify that the latent energy is closely related to the physical energy and final metrics.**
>
> **A2:** Furthermore, we employ the widely used xTB tool [1] to calculate the physical energy values of the molecular conformations obtained through different layers (i.e., given the $\mathbf{X}$ and $\mathbf{R}$ obtained by each layer, we can pass them through the trained decoder and obtain the corresponding molecular conformation) and analyze its correlation with the latent energy values obtained by Eq.17. In particular, taking the physical and latent energy obtained in the first layer as the references, we record the relative changes of the two kinds of energy w.r.t. number of layers and their correlations in the table below:
>
> | The number of layers                  | 5       | 10      | 15      | 20      | 25      | 30      |
> |--------------------------------------|---------|---------|---------|---------|---------|---------|
> | Relative Energy Value Change (kcal/mol) | -9.135  | -18.199 | -19.955 | -34.814 | -45.204 | -52.378 |
> | Relative Latent Energy Value Change     | -3.629  | -7.729  | -8.512  | -8.932  | -9.195  | -10.385 |
> | Pearson Correlation Coefficient (Energy VS Latent Energy) | 0.885 ± 0.033 |
> | Distance Correlation (Energy VS Latent Energy)           | 0.906 ± 0.018 |
>
> Here, a strong linear correlation is indicated by the Pearson correlation coefficient (0.885±0.033), while the slightly higher distance correlation (0.906±0.018) suggests additional nonlinear dependencies. **These results further validate the interpretability of our WGFormer --- minimizing the latent energy defined on the atoms' mixture model helps optimize the physically-meaningful energy of molecular conformation.**
>
> Finally, **as shown in Figure 5 of our paper, all evaluation metrics (D-MAE, D-RMSE, and C-RMSD) are improved steadily as the number of layers increases, having validated that minimizing the latent energy value can effectively improve the final metrics.**
>
> In general, the above results effectively verify the rationality and interpretability of our WGFormer, further supporting our theoretical claims proposed in Section 4 from the experimental point of view.
>
> Thank you once again for your valuable and insightful review, which has made our work more complete and convincing. We will add the above analytic experiments to the revised paper. We hope our responses resolve your concerns and make you more confident in supporting our work.
>
>
> Reference:
>
> [1] Bannwarth C, Ehlert S, Grimme S. GFN2-xTB—An accurate and broadly parametrized self-consistent tight-binding quantum chemical method with multipole electrostatics and density-dependent dispersion contributions. Journal of chemical theory and computation, 2019, 15(3): 1652-1671.

---

### Decision · Program_Chairs · 2025-05-01

**Decision:**

Accept (poster)

**Comment:**

The paper proposes a novel variant of SE(3) Transformer for the molecule 3D conformation transformation task (from a low-quality conformation to the ground-state conformation). The major design insight is a neural network architecture where each layer can be interpreted as one simulation step of the Wasserstein gradient flow (WGF) of a learnable functional (called "energy") of probability measure on an atom embedding space. The authors make an analogy to the physical energy function minimization process to find the ground-state conformation, in hope to enhance interpretability. The resulting model surpasses existing methods on common conformation prediction benchmarks.

All reviewers and I myself appreciate the significant novelty of the idea, which would inspire a new direction for designing architectures for molecular science. The results also seem promising.

The major concern is on the connection of the WGF of a learned energy functional to a physical energy minimization process. During the rebuttal, the authors visualized the correlation of the learned energy functional with the physical energy function (requested by Reviewer NScR) and additional results that have shared parameter (representing the same energy functional) over several layers (requested by Reviewer FN8S). These results provide more evidence on the WGF interpretation.

Nevertheless, I still sense a gap between the WGF interpretation and physical energy minimization. Conceptually, WGF minimizes a functional of a probability measure, while the physical energy is a function of the molecular conformation. The presented connection works by taking a conformation as a mixture of delta measures of atom embeddings. As at a delta measure, the optimal transport map may not exist, I would wonder if singularities would arise, e.g., if $\\frac{\\delta E}{\\delta \\mu}$ in Eq. (14) is not differentiable. What seems more natural to me following the energy function minimization perspective would be a ResNet whose residue update in each layer is the gradient of an energy function. I hope the authors could discuss the connection to this alternative and perform experiment comparison if necessary.